# GRADIENT DESCENT ASCENT FOR MIN-MAX PROBLEMS ON RIEMANNIAN MANIFOLDS

## ABSTRACT

In the paper, we study a class of useful non-convex minimax optimization problems on Riemanian manifolds and propose a class of Riemanian gradient descent ascent algorithms to solve these minimax problems. Specifically, we propose a new Riemannian gradient descent ascent (RGDA) algorithm for the deterministic minimax optimization. Moreover, we prove that the RGDA has a sample complexity of $O(\kappa^2\epsilon^{-2})$ for finding an $\epsilon$-stationary point of the nonconvex strongly-concave minimax problems, where $\kappa$ denotes the condition number. At the same time, we introduce a Riemannian stochastic gradient descent ascent (RSGDA) algorithm for the stochastic minimax optimization. In the theoretical analysis, we prove that the RSGDA can achieve a sample complexity of $O(\kappa^4\epsilon^{-4})$. To further reduce the sample complexity, we propose a novel momentum variance-reduced Riemannian stochastic gradient descent ascent (MVR-RSGDA) algorithm based on a new momentum variance-reduced technique of STORM. We prove that the MVR-RSGDA algorithm achieves a lower sample complexity of $\tilde{O}(\kappa^4\epsilon^{-3})$ without large batches, which reaches near the best known sample complexity for its Euclidean counterparts. Extensive experimental results on the robust deep neural networks training over Stiefel manifold demonstrate the efficiency of our proposed algorithms.

## 1 INTRODUCTION

In the paper, we study a class of useful non-convex minimax (*a.k.a.* min-max) problems on the Riemannian manifold $\mathcal{M}$ with the definition as:

$$\min_{x\in\mathcal{M}} \max_{y\in\mathcal{Y}} f(x,y), \tag{1}$$

where the function $f(x,y)$ is $\mu$-strongly concave in $y$ but possibly nonconvex in $x$. Here $\mathcal{Y} \subseteq \mathbb{R}^d$ is a convex and closed set. $f(\cdot, y) : \mathcal{M} \to \mathbb{R}$ for all $y \in \mathcal{Y}$ is a smooth but possibly nonconvex real-valued function on manifold $\mathcal{M}$, and $f(x, \cdot) : \mathcal{Y} \to \mathbb{R}$ for all $x \in \mathcal{M}$ a smooth and (strongly)-concave real-valued function. In this paper, we mainly focus on the stochastic minimax optimization problem $f(x,y) := \mathbb{E}_{\xi\sim\mathcal{D}}[f(x,y;\xi)]$, where $\xi$ is a random variable that follows an unknown distribution $\mathcal{D}$. In fact, the problem (1) is associated to many existing machine learning applications:

**1). Robust Training DNNs over Riemannian manifold.** Deep Neural Networks (DNNs) recently have been demonstrating exceptional performance on many machine learning applications. However, they are vulnerable to the adversarial example attacks, which show that a small perturbation in the data input can significantly change the output of DNNs. Thus, the security properties of DNNs have been widely studied. One of secured DNN research topics is to enhance the robustness of DNNs under the adversarial example attacks. To be more specific, given training data $\mathcal{D} := \{\xi_i = (a_i, b_i)\}_{i=1}^n$, where $a_i \in \mathbb{R}^d$ and $b_i \in \mathbb{R}$ represent the features and label of sample $\xi_i$ respectively. Each data sample $a_i$ can be corrupted by a universal small perturbation vector $y$ to generate an adversarial attack sample $a_i + y$, as in (Moosavi-Dezfooli et al., 2017; Chaubey et al., 2020). To make DNNs robust against adversarial attacks, one popular approach is to solve the following robust training problem:

$$\min_x \max_{y\in\mathcal{Y}} \frac{1}{n} \sum_{i=1}^n \ell(h(a_i + y; x), b_i), \tag{2}$$

where $y \in \mathbb{R}^d$ denotes a universal perturbation, and $x$ is the weight of the neural network; $h(\cdot;x)$ is the the deep neural network parameterized by $x$; and $\ell(\cdot)$ is the loss function. Here the constraint $\mathcal{Y} = \{y : \|y\| \leq \varepsilon\}$ indicates that the poisoned samples should not be too different from the original ones.

Recently, the orthonormality on weights of DNNs has gained much interest and has been found to be useful across different tasks such as person re-identification (Sun et al., 2017) and image classification (Xie et al., 2017). In fact, the orthonormality constraints improve the performances of DNNs (Li et al., 2020; Bansal et al., 2018), and reduce overfitting to improve generalization (Cogswell et al., 2015). At the same time, the orthonormality can stabilize the distribution of activations over layers within DNNs (Huang et al., 2018). Thus, we consider the following robust training problem over the Stiefel manifold $\mathcal{M}$:

$$\min_{x \in \mathcal{M}} \max_{y \in \mathcal{Y}} \frac{1}{n} \sum_{i=1}^{n} \ell(h(a_i + y; x), b_i). \tag{3}$$

When data are continuously coming, we can rewrite the problem (3) as follows:

$$\min_{x \in \mathcal{M}} \max_{y \in \mathcal{Y}} \mathbb{E}_{\xi}[f(x, y; \xi)], \tag{4}$$

where $f(x, y; \xi) = \ell(h(a + y; x), b)$ with $\xi = (a, b)$.

**2). Distributionally Robust Optimization over Riemannian manifold.** Distributionally robust optimization (DRO) (Chen et al., 2017; Rahimian & Mehrotra, 2019) is an effective method to deal with the noisy data, adversarial data, and imbalanced data. At the same time, the DRO in the Riemannian manifold setting is also widely applied in machine learning problems such as robust principal component analysis (PCA). To be more specific, given a set of data samples $\{\xi_i\}_{i=1}^{n}$, the DRO over Riemannian manifold $\mathcal{M}$ can be written as the following minimax problem:

$$\min_{x \in \mathcal{M}} \max_{\mathbf{p} \in \mathcal{S}} \left\{ \sum_{i=1}^{n} p_i \ell(x; \xi_i) - \|\mathbf{p} - \frac{\mathbf{1}}{n}\|^2 \right\}, \tag{5}$$

where $\mathbf{p} = (p_1, \cdots, p_n)$, $\mathcal{S} = \{\mathbf{p} \in \mathbb{R}^n : \sum_{i=1}^{n} p_i = 1, p_i \geq 0\}$. Here $\ell(x; \xi_i)$ denotes the loss function over Riemannian manifold $\mathcal{M}$, which applies to many machine learning problems such as PCA (Han & Gao, 2020a), dictionary learning (Sun et al., 2016), DNNs (Huang et al., 2018), structured low-rank matrix learning (Jawanpuria & Mishra, 2018), among others. For example, the task of PCA can be cast on a Grassmann manifold.

To the best of our knowledge, the existing explicitly minimax optimization methods such as gradient descent ascent method only focus on the minimax problems in Euclidean space. To fill this gap, in the paper, we propose a class of efficient Riemannian gradient descent ascent algorithms to solve the problem (1) via using general retraction and vector transport. When the problem (1) is deterministic, we propose a new deterministic Riemannian gradient descent ascent algorithm. When the problem (1) is stochastic, we propose two efficient stochastic Riemannian gradient descent ascent algorithms. Our main **contributions** can be summarized as follows:

1) We propose a novel Riemannian gradient descent ascent (RGDA) algorithm for the deterministic minimax optimization problem (1). We prove that the RGDA has a sample complexity of $O(\kappa^2 \epsilon^{-2})$ for finding an $\epsilon$-stationary point.

2) We also propose a new Riemannian stochastic gradient descent ascent (RSGDA) algorithm for the stochastic minimax optimization. In the theoretical analysis, we prove that the SRGDA has a sample complexity of $O(\kappa^4 \epsilon^{-4})$.

3) To further reduce the sample complexity, we introduce a novel momentum variance-reduced Riemannian stochastic gradient descent ascent (MVR-RSGDA) algorithm based on a new momentum variance-reduced technique of STORM (Cutkosky & Orabona, 2019). We prove the MVR-RSGDA achieves a lower sample complexity of $\tilde{O}(\kappa^4 \epsilon^{-3})$ (please see Table 1), which reaches near the best known sample complexity for its Euclidean counterparts.

4) Extensive experimental results on the robust DNN training over Stiefel manifold demonstrate the efficiency of our proposed algorithms.

Table 1: Convergence properties comparison of our algorithms for obtaining an $\epsilon$-stationary point of the min-max optimization problem (1). $\kappa$ denotes the condition number of function $f(x, \cdot)$.

| Problem | Algorithm | Learning Rate | Batch Size | Complexity |
|---|---|---|---|---|
| Deterministic | RGDA | Constant | – | $O(\kappa^2 \epsilon^{-2})$ |
| Stochastic | RSGDA | Constant | $O(\kappa^2 \epsilon^{-2})$ | $O(\kappa^4 \epsilon^{-4})$ |
| | MVR-RSGDA | Decrease | $O(1)$ | $\tilde{O}(\kappa^{9/2} \epsilon^{-3})$ |
| | MVR-RSGDA | Decrease | $O(\kappa)$ | $\tilde{O}(\kappa^4 \epsilon^{-3})$ |

## 2 RELATED WORKS

In this section, we briefly review the minimax optimization and Riemannian manifold optimization research works.

### 2.1 MINIMAX OPTIMIZATION

Minimax optimization recently has been widely applied in many machine learning problems such as adversarial training (Goodfellow et al., 2014; Liu et al., 2019), reinforcement learning (Zhang et al., 2019; 2020), and distribution learning (Razaviyayn et al., 2020). At the same time, many efficient min-max methods (Rafique et al., 2018; Lin et al., 2019; Nouiehed et al., 2019; Thekumparampil et al., 2019; Lin et al., 2020; Yang et al., 2020; Ostrovskii et al., 2020; Yan et al., 2020; Xu et al., 2020a; Luo et al., 2020; Xu et al., 2020b; Boţ & Böhm, 2020; Huang et al., 2020) have been proposed for solving these minimax optimization problems. For example, Thekumparampil et al. (2019) have proposed a class of efficient dual implicit accelerated gradient algorithms to solve smooth min-max optimization. Lin et al. (2019) have proposed a class of efficient gradient decent ascent methods for non-convex minimax optimization. Subsequently, accelerated first-order algorithms Lin et al. (2020) have been proposed for minimax optimization. Xu et al. (2020b) have proposed a unified single-loop alternating gradient projection algorithm for (non)convex-(non)concave minimax problems. Ostrovskii et al. (2020) have proposed an efficient algorithm for finding first-order Nash equilibria in nonconvex concave minimax problems. Xu et al. (2020a); Luo et al. (2020) have proposed a class of fast stochastic variance-reduced GDA algorithms to solve the stochastic minimax problems. More recently, Huang et al. (2020) have presented a class of new momentum-based first-order and zeroth-order descent ascent method for the nonconvex strongly concave minimax problems.

### 2.2 RIEMANNIAN MANIFOLD OPTIMIZATION

Riemannian manifold optimization methods have been widely applied in machine learning problems including dictionary learning (Sun et al., 2016), matrix factorization (Vandereycken, 2013), and DNNs (Huang et al., 2018). Many Riemannian optimization methods were recently proposed. *E.g.* Zhang & Sra (2016); Liu et al. (2017) have proposed some efficient first-order gradient methods for geodesically convex functions. Subsequently, Zhang et al. (2016) have presented fast stochastic variance-reduced methods to Riemannian manifold optimization. More recently, Sato et al. (2019) have proposed fast first-order gradient algorithms for Riemannian manifold optimization by using general retraction and vector transport. Subsequently, based on these retraction and vector transport, some fast Riemannian gradient-based methods (Zhang et al., 2018; Kasai et al., 2018; Zhou et al., 2019; Han & Gao, 2020a) have been proposed for non-convex optimization. Riemannian Adam-type algorithms (Kasai et al., 2019) were introduced for matrix manifold optimization. In addition, some algorithms (Ferreira et al., 2005; Li et al., 2009; Wang et al., 2010) have been studied for variational inequalities on Riemannian manifolds, which are the implicit min-max problems on Riemannian manifolds.

**Notations:** $\| \cdot \|$ denotes the $\ell_2$ norm for vectors and spectral norm for matrices. $\langle x, y \rangle$ denotes the inner product of two vectors $x$ and $y$. For function $f(x, y)$, $f(x, \cdot)$ denotes function *w.r.t.* the second variable with fixing $x$, and $f(\cdot, y)$ denotes function *w.r.t.* the first variable with fixing $y$. Given a convex closed set $\mathcal{Y}$, we define a projection operation on the set $\mathcal{Y}$ as $\mathcal{P}_{\mathcal{Y}}(y_0) = \arg\min_{y \in \mathcal{Y}} \frac{1}{2} \|y - y_0\|^2$. We denote $a = O(b)$ if $a \leq Cb$ for some constant $C > 0$, and the notation $\tilde{O}(\cdot)$ hides logarithmic terms. $I_d$ denotes the identity matrix with $d$ dimension. The operation $\bigoplus$ denotes the Whitney sum. Given $\mathcal{B}_t = \{\xi_t^i\}_{i=1}^B$ for any $t \geq 1$, let $\nabla f_{\mathcal{B}_t}(x, y) = \frac{1}{B} \sum_{i=1}^B \nabla f(x, y; \xi_t^i)$.

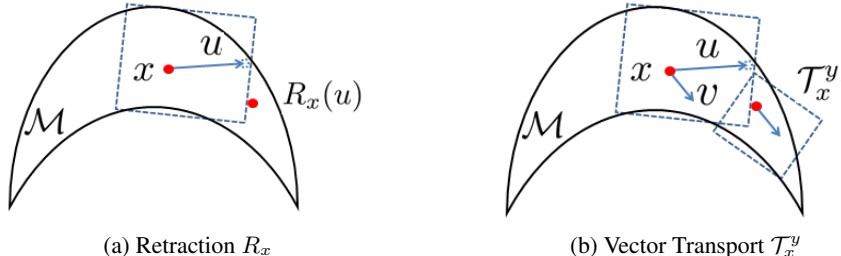

(a) Retraction $R_x$          (b) Vector Transport $\mathcal{T}_x^y$

Figure 1: Illustration of manifold operations.(a) A vector $u$ in $T_x\mathcal{M}$ is mapped to $R_x(u)$ in $\mathcal{M}$; (b) A vector $v$ in $T_x\mathcal{M}$ is transported to $T_y\mathcal{M}$ by $\mathcal{T}_x^y v$ (or $\mathcal{T}_u v$), where $y = R_x(u)$ and $u \in T_x\mathcal{M}$.

## 3    PRELIMINARIES

In this section, we first re-visit some basic information on the Riemannian manifold $\mathcal{M}$. In general, the manifold $\mathcal{M}$ is endowed with a smooth inner product $\langle \cdot, \cdot \rangle_x : T_x\mathcal{M} \times T_x\mathcal{M} \to \mathbb{R}$ on tangent space $T_x\mathcal{M}$ for every $x \in \mathcal{M}$. The induced norm $\| \cdot \|_x$ of a tangent vector in $T_x\mathcal{M}$ is associated with the Riemannian metric. We first define a retraction $R_x : T_x\mathcal{M} \to \mathcal{M}$ mapping tangent space $T_x\mathcal{M}$ onto $\mathcal{M}$ with a local rigidity condition that preserves the gradients at $x \in \mathcal{M}$ (please see Fig.1 (a)). The retraction $R_x$ satisfies all of the following: 1) $R_x(0) = x$, where $0 \in T_x\mathcal{M}$; 2) $\langle \nabla R_x(0), u \rangle_x = u$ for $u \in T_x\mathcal{M}$. In fact, exponential mapping $\text{Exp}_x$ is a special case of retraction $R_x$ that locally approximates the exponential mapping $\text{Exp}_x$ to the first order on the manifold.

Next, we define a vector transport $\mathcal{T} : T\mathcal{M} \bigoplus T\mathcal{M} \to T\mathcal{M}$ (please see Fig.1 (b)) that satisfies all of the following 1) $\mathcal{T}$ has an associated retraction $R$, i.e., for $x \in \mathcal{M}$ and $w, u \in T_x\mathcal{M}$, $\mathcal{T}_u w$ is a tangent vector at $R_x(w)$; 2) $\mathcal{T}_0 v = v$; 3) $\mathcal{T}_u(av + bw) = a\mathcal{T}_u v + b\mathcal{T}_u w$ for all $a, b \in \mathbb{R}$ a $u, v, w \in T\mathcal{M}$. Vector transport $\mathcal{T}_x^y v$ or equivalently $\mathcal{T}_u v$ with $y = R_x(u)$ transports $v \in T_x\mathcal{M}$ along the retraction curve defined by direction $u$. Here we focus on the isometric vector transport $\mathcal{T}_x^y$, which satisfies $\langle u, v \rangle_x = \langle \mathcal{T}_x^y u, \mathcal{T}_x^y v \rangle_y$ for all $u, v \in T_x\mathcal{M}$.

Let $\nabla f(x, y) = (\nabla_x f(x, y), \nabla_y f(x, y))$ denote the gradient over the Euclidean space, and let $\text{grad} f(x, y) = (\text{grad}_x f(x, y), \text{grad}_y f(x, y)) = \text{Proj}_{T_x\mathcal{M}}(\nabla f(x, y))$ denote the Riemannian gradient over tangent space $T_x\mathcal{M}$, where $\text{Proj}_{\mathcal{X}}(z) = \arg\min_{x \in \mathcal{X}} \|x - z\|$ is a projection operator. Based on the above definitions, we provide some standard assumptions about the problem (1). Although the problem (1) is non-convex, following (Von Neumann & Morgenstern, 2007), there exists a local solution or stationary point $(x^*, y^*)$ satisfies the Nash Equilibrium, i.e.,

$$f(x^*, y) \leq f(x^*, y^*) \leq f(x, y^*),$$

where $x^* \in \mathcal{X}$ and $y^* \in \mathcal{Y}$. Here $\mathcal{X} \subset \mathcal{M}$ is a neighbourhood around an optimal point $x^*$.

**Assumption 1.** *$\mathcal{X}$ is compact. Each component function $f(x, y)$ is twice continuously differentiable in both $x \in \mathcal{X}$ and $y \in \mathcal{Y}$, and there exist constants $L_{11}$, $L_{12}$, $L_{21}$ and $L_{22}$, such that for every $x, x_1, x_2 \in \mathcal{X}$ and $y, y_1, y_2 \in \mathcal{Y}$, we have*

$$\|grad_x f(x_1, y; \xi) - \mathcal{T}_{x_2}^{x_1} grad_x f(x_2, y; \xi)\| \leq L_{11}\|u\|,$$
$$\|grad_x f(x, y_1; \xi) - grad_x f(x, y_2; \xi)\| \leq L_{12}\|y_1 - y_2\|,$$
$$\|\nabla_y f(x_1, y; \xi) - \nabla_y f(x_2, y; \xi)\| \leq L_{21}\|u\|,$$
$$\|\nabla_y f(x, y_1; \xi) - \nabla_y f(x, y_2; \xi)\| \leq L_{22}\|y_1 - y_2\|,$$

*where $u \in T_{x_1}\mathcal{M}$ and $x_2 = R_{x_1}(u)$.*

Assumption 1 is commonly used in Riemannian optimization (Sato et al., 2019; Han & Gao, 2020a), and min-max optimization (Lin et al., 2019; Luo et al., 2020; Xu et al., 2020b). Here, the terms $L_{11}$, $L_{12}$ and $L_{21}$ implicitly contain the curvature information as in (Sato et al., 2019; Han & Gao, 2020a). Specifically, Assumption 1 implies the partial Riemannian gradient $\text{grad}_x f(\cdot, y; \xi)$ for all $y \in \mathcal{Y}$ is retraction $L_{11}$-Lipschitz continuous as in (Han & Gao, 2020a) and the partial gradient $\nabla_y f(x, \cdot; \xi)$ for all $x \in \mathcal{X}$ is $L_{22}$-Lipschitz continuous as in (Lin et al., 2019). Since $\|\text{grad}_x f(x, y_1; \xi) - \text{grad}_x f(x, y_2; \xi)\| = \|\text{Proj}_{T_x\mathcal{M}}(\nabla_x f(x, y_1; \xi)) - $

$\text{Proj}_{T_x\mathcal{M}}\big(\nabla_x f(x,y_2;\xi)\big)\| \leq \|\nabla_x f(x,y_1;\xi) - \nabla_x f(x,y_2;\xi)\| \leq L_{12}\|y_1 - y_2\|$, we can obtain $\|\text{grad}_x f(x,y_1;\xi) - \text{grad}_x f(x,y_2;\xi)\| \leq L_{12}\|y_1 - y_2\|$ by the $L_{12}$-Lipschitz continuous of $\nabla_x f(x,\cdot;\xi)$ for all $x \in \mathcal{X}$. Let the partial Riemannian gradient $\text{grad}_y f(\cdot,y;\xi)$ for all $y \in \mathcal{Y}$ be retraction $\tilde{L}_{21}$-Lipschitz, i.e., $\|\text{grad}_y f(x_1,y;\xi) - \mathcal{T}_{x_2}^{x_1}\text{grad}_y f(x_2,y;\xi)\| \leq \tilde{L}_{21}\|u\|$. Since $\|\text{grad}_y f(x_1,y;\xi) - \mathcal{T}_{x_2}^{x_1}\text{grad}_y f(x_2,y;\xi)\| = \|\text{Proj}_{T_x\mathcal{M}}\big(\nabla_y f(x_1,y;\xi)\big) - \mathcal{T}_{x_2}^{x_1}\text{Proj}_{T_x\mathcal{M}}\big(\nabla_y f(x_2,y;\xi)\big)\| \leq \|\nabla_y f(x_1,y;\xi) - \nabla_y f(x_2,y;\xi)\| \leq L_{21}\|u\|$, we have $L_{21} \geq \tilde{L}_{21}$.

For the deterministic problem, let $f(x,y)$ instead of $f(x,y;\xi)$ in Assumption 1. Since $f(x,y)$ is strongly concave in $y \in \mathcal{Y}$, there exists a unique solution to the problem $\max_{y\in\mathcal{Y}} f(x,y)$ for any $x$. We define the function $\Phi(x) = \max_{y\in\mathcal{Y}} f(x,y)$ and $y^*(x) = \arg\max_{y\in\mathcal{Y}} f(x,y)$.

**Assumption 2.** *The function $\Phi(x)$ is retraction L-smooth. There exists a constant $L > 0$, for all $x \in \mathcal{X}, z = R_x(u)$ with $u \in T_x\mathcal{M}$, such that*

$$\Phi(z) \leq \Phi(x) + \langle grad\Phi(x), u\rangle + \frac{L}{2}\|u\|^2. \tag{6}$$

**Assumption 3.** *The objective function $f(x,y)$ is $\mu$-strongly concave w.r.t $y$, i.e., for any $x \in \mathcal{M}$*

$$f(x,y_1) \leq f(x,y_2) + \langle\nabla_y f(x,y_2), y_1 - y_2\rangle - \frac{\mu}{2}\|y_1 - y_2\|^2, \ \forall y_1, y_2 \in \mathcal{Y}. \tag{7}$$

**Assumption 4.** *The function $\Phi(x)$ is bounded from below in $\mathcal{M}$, i.e., $\Phi^* = \inf_{x\in\mathcal{M}} \Phi(x)$.*

**Assumption 5.** *The variance of stochastic gradient is bounded, i.e., there exists a constant $\sigma_1 > 0$ such that for all $x$, it follows $\mathbb{E}_\xi\|grad_x f(x,y;\xi) - grad_x f(x,y)\|^2 \leq \sigma_1^2$; There exists a constant $\sigma_2 > 0$ such that for all $y$, it follows $\mathbb{E}_\xi\|\nabla_y f(x,y;\xi) - \nabla_y f(x,y)\|^2 \leq \sigma_2^2$. We also define $\sigma = \max\{\sigma_1, \sigma_2\}$.*

Assumption 2 imposes the retraction smooth of function $\Phi(x)$, as in Sato et al. (2019); Han & Gao (2020b;a). Assumption 3 imposes the strongly concave of $f(x,y)$ on variable $y$, as in (Lin et al., 2019; Luo et al., 2020). Assumption 4 guarantees the feasibility of the nonconvex-strongly-concave problems, as in (Lin et al., 2019; Luo et al., 2020). Assumption 5 imposes the bounded variance of stochastic (Riemannian) gradients, which is commonly used in the stochastic optimization (Han & Gao, 2020b; Lin et al., 2019; Luo et al., 2020).

## 4 RIEMANIAN GRADIENT DESCENT ASCENT

In the section, we propose a class of Riemannian gradient descent ascent algorithm to solve the deterministic and stochastic minimax optimization problem (1), respectively.

### 4.1 RGDA AND RSGDA ALGORITHMS

In this subsection, we propose an efficient Riemannian gradient descent ascent (RGDA) algorithm to solve the deterministic min-max problem (1). At the same time, we propose a standard Riemannian stochastic gradient descent ascent (RSGDA) algorithm to solve the stochastic min-max problem (1). Algorithm 1 summarizes the algorithmic framework of our RGDA and RSGDA algorithms.

At the step 5 of Algorithm 1, we apply the retraction operator to ensure the variable $x_t$ for all $t \geq 1$ in the manifold $\mathcal{M}$. At the step 6 of Algorithm 1, we use $0 < \eta_t \leq 1$ to ensure the variable $y_t$ for all $t \geq 1$ in the convex constraint $\mathcal{Y}$.

Here we define a reasonable metric to measure the convergence:

$$\mathcal{H}_t = \|\text{grad}\Phi(x_t)\| + \tilde{L}\|y_t - y^*(x_t)\|, \tag{10}$$

where $\tilde{L} = \max(1, L_{11}, L_{12}, L_{21}, L_{22})$, and the first term of $\mathcal{H}_t$ measures convergence of the iteration solutions $\{x_t\}_{t=1}^T$, and the last term measures convergence of the iteration solutions $\{y_t\}_{t=1}^T$. Since the function $f(x,y)$ is strongly concave in $y \in \mathcal{Y}$, there exists a unique solution $y^*(x)$ to the problem $\max_{y\in\mathcal{Y}} f(x,y)$ for any $x \in \mathcal{M}$. Thus, we apply the standard metric $\|y_t - y^*(x_t)\|$ to measure convergence of the parameter $y$. Given $y = y^*(x_t)$, we use the standard metric $\|\text{grad}\Phi(x_t)\| = \|\text{grad}_x f(x_t, y^*(x_t))\|$ to measure convergence of the parameter $x$. Note that we use the coefficient $\tilde{L}$ to balance the scale of metrics of the variable $x$ and the variable $y$.

---

**Algorithm 1** RGDA and RSGDA Algorithms for Min-Max Optimization

---

1: **Input:** $T$, parameters $\{\gamma, \lambda, \eta_t\}_{t=1}^T$, mini-batch size $B$, and initial input $x_1 \in \mathcal{M}, y_1 \in \mathcal{Y}$;
2: **for** $t = 1, 2, \ldots, T$ **do**
3:     **(RGDA)** Compute deterministic gradients

$$v_t = \text{grad}_x f(x_t, y_t), \ w_t = \nabla_y f(x_t, y_t); \tag{8}$$

4:     **(RSGDA)** Draw $B$ i.i.d. samples $\{\xi_t^i\}_{i=1}^B$, then compute stochastic gradients

$$v_t = \frac{1}{B}\sum_{i=1}^B \text{grad}_x f(x_t, y_t; \xi_t^i), \ w_t = \frac{1}{B}\sum_{i=1}^B \nabla_y f(x_t, y_t; \xi_t^i); \tag{9}$$

5:     Update: $x_{t+1} = R_{x_t}(-\gamma \eta_t v_t)$;
6:     Update: $\tilde{y}_{t+1} = \mathcal{P}_{\mathcal{Y}}(y_t + \lambda w_t)$ and $y_{t+1} = y_t + \eta_t(\tilde{y}_{t+1} - y_t)$;
7: **end for**
8: **Output:** $x_\zeta$ and $y_\zeta$ chosen uniformly random from $\{x_t, y_t\}_{t=1}^T$.

---

**Algorithm 2** MVR-RSGDA Algorithm for Min-Max Optimization

---

1: **Input:** $T$, parameters $\{\gamma, \lambda, b, m, c_1, c_2\}$ and initial input $x_1 \in \mathcal{M}$ and $y_1 \in \mathcal{Y}$;
2: Draw $B$ i.i.d. samples $\mathcal{B}_1 = \{\xi_1^i\}_{i=1}^B$, then compute $v_1 = \text{grad}_x f_{\mathcal{B}_1}(x_1, y_1)$ and $w_1 = \nabla_y f_{\mathcal{B}_1}(x_1, y_1)$;
3: **for** $t = 1, 2, \ldots, T$ **do**
4:     Compute $\eta_t = \frac{b}{(m+t)^{1/3}}$;
5:     Update: $x_{t+1} = R_{x_t}(-\gamma \eta_t v_t)$;
6:     Update: $\tilde{y}_{t+1} = \mathcal{P}_{\mathcal{Y}}(y_t + \lambda w_t)$ and $y_{t+1} = y_t + \eta_t(\tilde{y}_{t+1} - y_t)$;
7:     Compute $\alpha_{t+1} = c_1 \eta_t^2$ and $\beta_{t+1} = c_2 \eta_t^2$;
8:     Draw $B$ i.i.d. samples $\mathcal{B}_{t+1} = \{\xi_{t+1}^i\}_{i=1}^B$, then compute

$$v_{t+1} = \text{grad}_x f_{\mathcal{B}_{t+1}}(x_{t+1}, y_{t+1}) + (1 - \alpha_{t+1})\mathcal{T}_{x_t}^{x_{t+1}}[v_t - \text{grad}_x f_{\mathcal{B}_{t+1}}(x_t, y_t)], \tag{12}$$

$$w_{t+1} = \nabla_y f_{\mathcal{B}_{t+1}}(x_{t+1}, y_{t+1}) + (1 - \beta_{t+1})[w_t - \nabla_y f_{\mathcal{B}_{t+1}}(x_t, y_t)]; \tag{13}$$

9: **end for**
10: **Output:** $x_\zeta$ and $y_\zeta$ chosen uniformly random from $\{x_t, y_t\}_{t=1}^T$.

---

### 4.2 MVR-RSGDA ALGORITHM

In this subsection, we propose a novel momentum variance-reduced stochastic Riemannian gradient descent ascent (MVR-RSGDA) algorithm to solve the stochastic min-max problem (1), which builds on the momentum-based variance reduction technique of STORM (Cutkosky & Orabona, 2019). Algorithm 2 describes the algorithmic framework of MVR-RSGDA method.

In Algorithm 2, we use the momentum-based variance-reduced technique of STORM to update stochastic Riemannian gradient $v_t$:

$$\begin{aligned}
v_{t+1} = \alpha_{t+1} \underbrace{\text{grad}_x f_{\mathcal{B}_{t+1}}(x_{t+1}, y_{t+1})}_{\text{SGD}} \\
+ (1 - \alpha_{t+1}) \underbrace{\left(\text{grad}_x f_{\mathcal{B}_{t+1}}(x_{t+1}, y_{t+1}) - \mathcal{T}_{x_t}^{x_{t+1}}\left(\text{grad}_x f_{\mathcal{B}_{t+1}}(x_t, y_t) - v_t\right)\right)}_{\text{SPIDER}} \\
= \text{grad}_x f_{\mathcal{B}_{t+1}}(x_{t+1}, y_{t+1}) + (1 - \alpha_{t+1})\mathcal{T}_{x_t}^{x_{t+1}}\left(v_t - \text{grad}_x f_{\mathcal{B}_{t+1}}(x_t, y_t)\right),
\end{aligned} \tag{11}$$

where $\alpha_{t+1} \in (0, 1]$. When $\alpha_{t+1} = 1$, $v_t$ will degenerate a vanilla stochastic Riemannian gradient; When $\alpha_{t+1} = 0$, $v_t$ will degenerate a stochastic Riemannian gradient based on variance-reduced technique of SPIDER (Nguyen et al., 2017; Fang et al., 2018). Similarly, we use this momentum-based variance-reduced technique to estimate the stochastic gradient $w_t$.

## 5 CONVERGENCE ANALYSIS

In this section, we study the convergence properties of our RGDA, RSGDA, and MVR-RSGDA algorithms under some mild conditions. For notational simplicity, let $\tilde{L} = \max(1, L_{11}, L_{12}, L_{21}, L_{22})$ and $\kappa = L_{21}/\mu$ denote the number condition of function $f(x, y)$. We first give a useful lemma.

**Lemma 1.** *Under the assumptions in §3, the gradient of function $\Phi(x) = \max_{y \in \mathcal{Y}} f(x, y)$ is retraction G-Lipschitz, and the mapping or function $y^*(x) = \arg\max_{y \in \mathcal{Y}} f(x, y)$ is retraction $\kappa$-Lipschitz. Given any $x_1, x_2 = R_{x_1}(u) \in \mathcal{X} \subset \mathcal{M}$ and $u \in T_{x_1}\mathcal{M}$, we have:*

$$\|grad\Phi(x_1) - \mathcal{T}_{x_2}^{x_1} grad\Phi(x_2)\| \le G\|u\|, \quad \|y^*(x_1) - y^*(x_2)\| \le \kappa\|u\|, \quad (14)$$

*where $G = \kappa L_{12} + L_{11}$ and $\kappa = L_{21}/\mu$.*

### 5.1 CONVERGENCE ANALYSIS OF BOTH THE RGDA AND RSGDA ALGORITHMS

In the subsection, we study the convergence properties of deterministic RGDA and stochastic RSGDA algorithms. The related proofs of RGDA and RSGDA are provided in Appendix A.1.

**Theorem 1.** *Suppose the sequence $\{x_t, y_t\}_{t=1}^T$ is generated from Algorithm 1 by using **deterministic** gradients. Given $\eta = \eta_t$ for all $t \ge 1$, $0 < \eta \le \min(1, \frac{1}{2\gamma L})$, $0 < \lambda \le \frac{1}{6\tilde{L}}$ and $0 < \gamma \le \frac{\mu\lambda}{10\tilde{L}\kappa}$, we have*

$$\frac{1}{T}\sum_{t=1}^T \left[\|grad\Phi(x_t)\| + \tilde{L}\|y_t - y^*(x_t)\|\right] \le \frac{2\sqrt{\Phi(x_1) - \Phi^*}}{\sqrt{\gamma\eta T}}. \quad (15)$$

**Remark 1.** *Since $0 < \eta \le \min(1, \frac{1}{2\gamma L})$ and $0 < \gamma \le \frac{\mu\lambda}{10\tilde{L}\kappa}$, we have $0 < \eta\gamma \le \min(\frac{\mu\lambda}{10\tilde{L}\kappa}, \frac{1}{2L})$. Let $\eta\gamma = \min(\frac{\mu\lambda}{10\tilde{L}\kappa}, \frac{1}{2L})$, we have $\eta\gamma = O(\frac{1}{\kappa^2})$. The RGDA algorithm has convergence rate of $O\big(\frac{\kappa}{T^{1/2}}\big)$. By $\frac{\kappa}{T^{1/2}} \le \epsilon$, i.e., $\mathbb{E}[\mathcal{H}_\zeta] \le \epsilon$, we choose $T \ge \kappa^2\epsilon^{-2}$. In the deterministic RGDA Algorithm, we need one sample to estimate the gradients $v_t$ and $w_t$ at each iteration, and need $T$ iterations. Thus, the RGDA reaches a sample complexity of $T = O(\kappa^2\epsilon^{-2})$ for finding an $\epsilon$-stationary point.*

**Theorem 2.** *Suppose the sequence $\{x_t, y_t\}_{t=1}^T$ is generated from Algorithm 1 by using **stochastic** gradients. Given $\eta = \eta_t$ for all $t \ge 1$, $0 < \eta \le \min(1, \frac{1}{2\gamma L})$, $0 < \lambda \le \frac{1}{6\tilde{L}}$ and $0 < \gamma \le \frac{\mu\lambda}{10\tilde{L}\kappa}$, we have*

$$\frac{1}{T}\sum_{t=1}^T \mathbb{E}\left[\|grad\Phi(x_t)\| + \tilde{L}\|y_t - y^*(x_t)\|\right] \le \frac{2\sqrt{\Phi(x_1) - \Phi^*}}{\sqrt{\gamma\eta T}} + \frac{\sqrt{2}\sigma}{\sqrt{B}} + \frac{5\sqrt{2}\tilde{L}\sigma}{\sqrt{B}\mu}. \quad (16)$$

**Remark 2.** *Since $0 < \eta \le \min(1, \frac{1}{2\gamma L})$ and $0 < \gamma \le \frac{\mu\lambda}{10\tilde{L}\kappa}$, we have $0 < \eta\gamma \le \min(\frac{\mu\lambda}{10\tilde{L}\kappa}, \frac{1}{2L})$. Let $\eta\gamma = \min(\frac{\mu\lambda}{10\tilde{L}\kappa}, \frac{1}{2L})$, we have $\eta\gamma = O(\frac{1}{\kappa^2})$. Let $B = T$, the RSGDA algorithm has convergence rate of $O\big(\frac{\kappa}{T^{1/2}}\big)$. By $\frac{\kappa}{T^{1/2}} \le \epsilon$, i.e., $\mathbb{E}[\mathcal{H}_\zeta] \le \epsilon$, we choose $T \ge \kappa^2\epsilon^{-2}$. In the stochastic RSGDA Algorithm, we need $B$ samples to estimate the gradients $v_t$ and $w_t$ at each iteration, and need $T$ iterations. Thus, the RSGDA reaches a sample complexity of $BT = O(\kappa^4\epsilon^{-4})$ for finding an $\epsilon$-stationary point.*

### 5.2 CONVERGENCE ANALYSIS OF THE MVR-RSGDA ALGORITHM

In the subsection, we provide the convergence properties of the MVR-RSGDA algorithm. The related proofs of MVR-RSGDA are provided in Appendix A.2.

**Theorem 3.** *Suppose the sequence $\{x_t, y_t\}_{t=1}^T$ is generated from Algorithm 2. Given $y_1 = y^*(x_1)$, $c_1 \ge \frac{2}{3b^3} + 2\lambda\mu$, $c_2 \ge \frac{2}{3b^3} + \frac{50\lambda\tilde{L}^2}{\mu}$, $b > 0$, $m \ge \max\big(2, (\tilde{c}b)^3\big)$, $0 < \gamma \le \frac{\mu\lambda}{2\kappa\tilde{L}\sqrt{25+4\mu\lambda}}$ and $0 < \lambda \le \frac{1}{6\tilde{L}}$, we have*

$$\frac{1}{T}\sum_{t=1}^T \mathbb{E}\left[\|grad\Phi(x_t)\| + \tilde{L}\|y_t - y^*(x_t)\|\right] \le \frac{\sqrt{2M'}m^{1/6}}{T^{1/2}} + \frac{\sqrt{2M'}}{T^{1/3}}, \quad (17)$$

*where $\tilde{c} = \max(1, c_1, c_2, 2\gamma L)$ and $M' = \frac{2(\Phi(x_1) - \Phi^*)}{\gamma b} + \frac{2\sigma^2}{B\lambda\mu\eta_0 b} + \frac{2(c_1^2 + c_2^2)\sigma^2 b^2}{B\lambda\mu}\ln(m + T)$.*

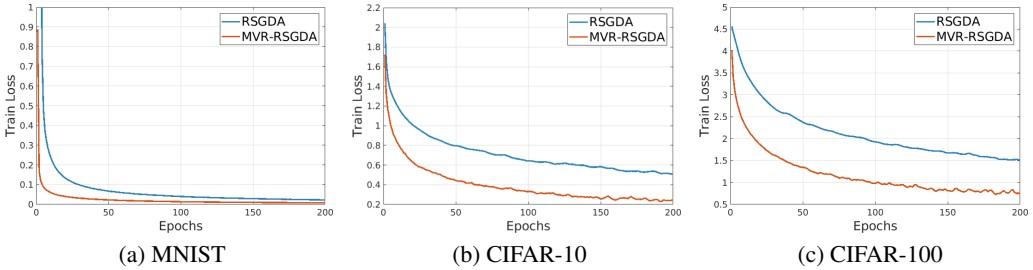

Figure 2: Training loss of robust training DNNs with orthogonality regularization on weights.

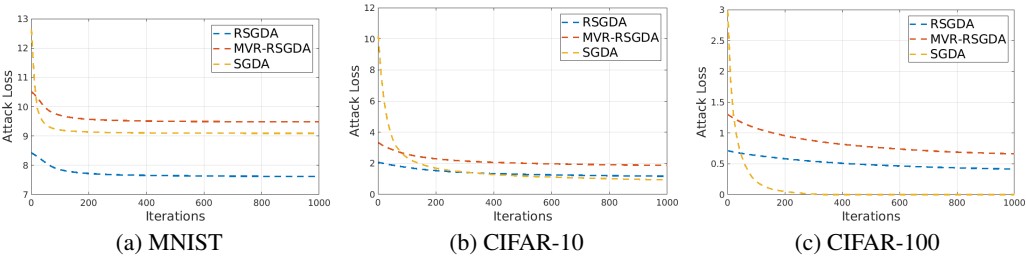

Figure 3: Attack loss when using uniform attack on DNNs trained by SGDA, RSGDA and MVR-RSGDA.

**Remark 3.** *Let $c_1 = \frac{2}{3b^3} + 2\lambda\mu$, $c_2 = \frac{2}{3b^3} + \frac{50\lambda\tilde{L}^2}{\mu}$, $\lambda = \frac{1}{6\tilde{L}}$, $\gamma = \frac{\mu\lambda}{2\kappa\tilde{L}\sqrt{25+4\mu\lambda}}$ and $\eta_0 = \frac{b}{m^{1/3}}$. It is easy verified that $\gamma = O(\frac{1}{\kappa^2})$, $\lambda = O(1)$, $\lambda\mu = O(\frac{1}{\kappa})$, $c_1 = O(1)$, $c_2 = O(\kappa)$, $m = O(\kappa^3)$ and $\eta_0 = O(\frac{1}{\kappa})$. Without loss of generality, let $T \geq m = O(\kappa^3)$, we have $M' = O(\kappa^2 + \frac{\kappa^2}{B} + \frac{\kappa^3}{B}\ln(T))$. When $B = \kappa$, we have $M' = O(\kappa^2\ln(T))$. Thus, the MVR-RSGDA algorithm has a convergence rate of $\tilde{O}(\frac{\kappa}{T^{1/3}})$. By $\frac{\kappa}{T^{1/3}} \leq \epsilon$, i.e., $\mathbb{E}[\mathcal{H}_\zeta] \leq \epsilon$, we choose $T \geq \kappa^3\epsilon^{-3}$. In Algorithm 2, we require $B$ samples to estimate the stochastic gradients $v_t$ and $w_t$ at each iteration, and need $T$ iterations. Thus, the MVR-RSGDA has a sample complexity of $BT = \tilde{O}(\kappa^4\epsilon^{-3})$ for finding an $\epsilon$-stationary point of the problem (1). Similarly, when $B = 1$, the MVR-RSGDA algorithm has a convergence rate of $\tilde{O}(\frac{\kappa^{3/2}}{T^{1/3}})$, and has a sample complexity of $BT = \tilde{O}(\kappa^{9/2}\epsilon^{-3})$ for finding an $\epsilon$-stationary point.*

**Remark 4.** *In the about theoretical analysis, we only assume the convexity of constraint set $\mathcal{Y}$, while Lin et al. (2019) not only assume the convexity of set $\mathcal{Y}$, but also assume and use its bounded (Please see Assumption 4.2 in (Lin et al., 2019)). Clearly, our assumption is milder than (Lin et al., 2019). When there does not exist a constraint set on parameter $y$, i.e., $\mathcal{Y} = R^d$, our algorithms and theoretical results still work, while Lin et al. (2019) can't work.*

## 6 EXPERIMENTS

In this section, we conduct the deep neural network (DNN) robust training over the Stiefel manifold $\text{St}(r, d) = \{W \in \mathbb{R}^{d \times r} : W^T W = I_r\}$ to evaluate the performance of our algorithms. In the experiment, we use MNIST, CIFAR-10, and CIFAR-100 datasets to train the model ( More experimental results on SVHN, STL10, and FashionMNIST datasets are provided in the Appendix B ). Considering the sample size is large in these datasets, we only compare the proposed stochastic algorithms (RSGDA and MVR-RSGDA) in the experiments. Here, we use the SGDA algorithm (Lin et al., 2019) as a baseline, which does not apply the orthogonal regularization in the DNN robust training.

### 6.1 EXPERIMENTAL SETTING

Given a deep neural network $h(\cdot; x)$ parameterized by $x$ as shown in the above problem (2), the weights of $l$-th layer is $x_i \in \text{St}(n_{\text{in}}^l, n_{\text{out}}^l)$, where $\text{St}(n_{\text{in}}^l, n_{\text{out}}^l)$ is the Stiefel manifold of $l$-th layer.

Table 2: Test accuracy against nature images and uniform attack for MNIST, CIFAR-10, and CIFAR-100 datasets.

| Methods | Eval. Against | MNIST | CIFAR-10 | CIFAR-100 |
|---------|---------------|-------|----------|-----------|
| RSGDA | Nat. Images | 98.92% | 62.87% | 29.92% |
| | Uniform Attack | 98.58% | 61.45% | 31.14% |
| MVR-RSGDA | Nat. Images | **99.15%** | 67.45% | **32.92%** |
| | Uniform Attack | **99.03%** | **68.56%** | 34.69% |
| SGDA | Nat. Images | 98.96% | **76.75%** | **43.41%** |
| | Uniform Attack | 98.59% | 55.68% | 27.81% |

For the weights in dense layers, $n_{\text{in}}^l, n_{\text{out}}^l$ are the number of inputs and outputs neurons. For the weights in convolution layers, $n_{\text{in}}^l$ is the number of input channels, $n_{\text{out}}^l$ is the product of the number of output channels and kernel sizes. Note that the trainable parameters from other components (e.g. batchnorm) are not in Stiefel manifold.

For both RSGDA and MVR-RSGDA algorithms, we set $\{\gamma, \lambda\}$ to $\{1.0, 0.1\}$. We further set $\{b, m, c_1, c_2\}$ to 0.5, 8, 512, 512 for MVR-RSGDA. $\eta$ in RSGDA is set to 0.01. For both algorithms, the mini-batch size is set to 512. We set $\epsilon$ for $y$ as 0.05 and 0.03 for the MNIST dataset and CIFAR-10/100 datasets. The above settings are the same for all datasets. An 8-layer (5 convolution layers and 3 dense layers) deep neural network is used in all experiments. All codes are implemented with McTorch (Meghwanshi et al., 2018) which is based on PyTorch (Paszke et al., 2019).

## 6.2 EXPERIMENTAL RESULTS

The training loss plots of the robust training problem in the above Eq. (2) are shown in Fig. 2. From the figure, we can see that MVR-RSGDA enjoys a faster convergence speed compared to the baseline RSGDA. It's also clear that when the dataset becomes complicate (from MNIST to CIFAR-10/100), the advantage of MVR-RSGDA becomes larger.

When it comes to robust training, the training loss is not enough to identify which algorithm is better. We also use a variant of uniform perturbation to attack the model trained by our algorithms. We follow the design of uniform attack in previous works (Moosavi-Dezfooli et al., 2017; Chaubey et al., 2020), and the detail uniform attack objective is shown below:

$$\min_{y \in \mathcal{Y}} \frac{1}{n} \sum_{i=1}^{n} \max\left(h_{b_i}(y + a_i) - \max_{j \neq b_i} h_j(y + a_i), 0\right), \quad \text{s.t. } \mathcal{Y} = \{\|y\|_\infty \leq \varepsilon\}$$

where $h_j$ is the $j$-th logit of the output from the deep neural network, and $y$ here is a uniform permutation added for all inputs. In practice, we sample a mini-batch with 512 samples at each iteration. The optimization of the uniform permutation lasts for 1000 iterations for all settings. The attack loss is presented in Fig 3. The attack loss for the model trained by MVR-RSGDA is higher compared to both RSGDA and SGDA, which indicates the model trained by MVR-RSGDA is harder to attack and thus more robust. The test accuracy with natural image and uniform attack is shown in Tab. 2, which also suggests the advantage of MVR-RSGDA. More results are provided in Appendix B.

## 7 CONCLUSION

In the paper, we investigated a class of useful min-max optimization problems on the Riemanian manifold. We proposed a class of novel efficient Riemanian gradient descent ascent algorithms to solve these minimax problems, and studied the convergence properties of the proposed algorithms. For example, we proved that our new MVR-RSGDA algorithm achieves a sample complexity of $\tilde{O}(\kappa^4 \epsilon^{-3})$ without large batches, which reaches near the best known sample complexity for its Euclidean counterparts.

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

## A  APPENDIX

In this section, we provide the detailed convergence analysis of our algorithms. We first review some useful lemmas.

**Lemma 2.** *(Nesterov, 2018) Assume that $f(x)$ is a differentiable convex function and $\mathcal{X}$ is a convex set. $x^* \in \mathcal{X}$ is the solution of the constrained problem $\min_{x \in \mathcal{X}} f(x)$, if*

$$\langle \nabla f(x^*), x - x^* \rangle \geq 0, \; x \in \mathcal{X}. \tag{18}$$

**Lemma 3.** *(Nesterov, 2018) Assume the function $f(x)$ is $L$-smooth, i.e., $\|\nabla f(x) - \nabla f(y)\| \leq L\|x - y\|$, and then the following inequality holds*

$$|f(y) - f(x) - \nabla f(x)^T (y - x)| \leq \frac{L}{2}\|x - y\|^2. \tag{19}$$

Next, based on the above assumptions and Lemmas, we gives some useful lemmas:

**Lemma 4.** *The gradient of function $\Phi(x) = \max_{y \in \mathcal{Y}} f(x, y)$ is retraction $G$-Lipschitz, and the mapping or function $y^*(x) = \arg\max_{y \in \mathcal{Y}} f(x, y)$ is retraction $\kappa$-Lipschitz. Given any $x_1, x_2 = R_{x_1}(u) \in \mathcal{X} \subset \mathcal{M}$ and $u \in T_{x_1}\mathcal{M}$, we have*

$$\|grad\Phi(x_1) - \mathcal{T}_{x_2}^{x_1} grad\Phi(x_2)\| \leq G\|u\|,$$
$$\|y^*(x_1) - y^*(x_2)\| \leq \kappa\|u\|,$$

*where $G = \kappa L_{12} + L_{11}$ and $\kappa = L_{21}/\mu$, and vector transport $\mathcal{T}_{x_2}^{x_1}$ transport the tangent space of $x_1$ to that of $x_2$.*

*Proof.* Given any $x_1, x_2 = R_{x_1}(u) \in \mathcal{X}$ and $u \in T_{x_1}\mathcal{M}$, define $y^*(x_1) = \arg\max_{y \in \mathcal{Y}} f(x_1, y)$ and $y^*(x_2) = \arg\max_{y \in \mathcal{Y}} f(x_2, y)$, by the above Lemma 2, we have

$$(y - y^*(x_1))^T \nabla_y f(x_1, y^*(x_1)) \leq 0, \quad \forall y \in \mathcal{Y} \tag{20}$$
$$(y - y^*(x_2))^T \nabla_y f(x_2, y^*(x_2)) \leq 0, \quad \forall y \in \mathcal{Y}. \tag{21}$$

Let $y = y^*(x_2)$ in the inequality (20) and $y = y^*(x_1)$ in the inequality (21), then summing these inequalities, we have

$$(y^*(x_2) - y^*(x_1))^T \big(\nabla_y f(x_1, y^*(x_1)) - \nabla_y f(x_2, y^*(x_2))\big) \leq 0. \tag{22}$$

Since the function $f(x_1, \cdot)$ is $\mu$-strongly concave, we have

$$f(x_1, y^*(x_1)) \leq f(x_1, y^*(x_2)) + (\nabla_y f(x_1, y^*(x_2)))^T (y^*(x_1) - y^*(x_2)) - \frac{\mu}{2}\|y^*(x_1) - y^*(x_2)\|^2, \tag{23}$$

$$f(x_1, y^*(x_2)) \leq f(x_1, y^*(x_1)) + (\nabla_y f(x_1, y^*(x_1)))^T (y^*(x_2) - y^*(x_1)) - \frac{\mu}{2}\|y^*(x_1) - y^*(x_2)\|^2. \tag{24}$$

Combining the inequalities (23) with (24), we obtain

$$(y^*(x_2) - y^*(x_1))^T \big(\nabla_y f(x_1, y^*(x_2)) - \nabla_y f(x_1, y^*(x_1))\big) + \mu\|y^*(x_1) - y^*(x_2)\|^2 \leq 0. \tag{25}$$

By plugging the inequalities (22) into (25), we have

$$\mu\|y^*(x_1) - y^*(x_2)\|^2 \leq (y^*(x_2) - y^*(x_1))^T \big(\nabla_y f(x_2, y^*(x_2)) - \nabla_y f(x_1, y^*(x_2))\big)$$
$$\leq \|y^*(x_2) - y^*(x_1)\|\|\nabla_y f(x_2, y^*(x_2)) - \nabla_y f(x_1, y^*(x_2))\|$$
$$\leq L_{21}\|u\|\|y^*(x_2) - y^*(x_1)\|, \tag{26}$$

where the last inequality is due to Assumption 1. Thus, we have

$$\|y^*(x_1) - y^*(x_2)\| \leq \kappa\|u\|, \tag{27}$$

where $\kappa = L_{21}/\mu$ and $x_2 = R_{x_1}(u)$, $u \in T_{x_1}\mathcal{M}$.

Since $\Phi(x) = f(x, y^*(x))$, we have $\text{grad}\Phi(x) = \text{grad}_x f(x, y^*(x))$. Then we have

$\|\text{grad}\Phi(x_1) - \mathcal{T}_{x_2}^{x_1}\text{grad}\Phi(x_2)\|$

$= \|\text{grad}_x f(x_1, y^*(x_1)) - \mathcal{T}_{x_2}^{x_1}\text{grad}_x f(x_2, y^*(x_2))\|$

$\leq \|\text{grad}_x f(x_1, y^*(x_1)) - \text{grad}_x f(x_1, y^*(x_2))\| + \|\text{grad}_x f(x_1, y^*(x_2)) - \mathcal{T}_{x_2}^{x_1}\text{grad}_x f(x_2, y^*(x_2))\|$

$\leq L_{12}\|y^*(x_1) - y^*(x_2)\| + L_{11}\|u\|$

$\leq (\kappa L_{12} + L_{11})\|u\|,$  (28)

where $u \in T_{x_1}\mathcal{M}$.

$\square$

**Lemma 5.** *Suppose the sequence $\{x_t, y_t\}_{t=1}^T$ is generated from Algorithm 1 or 2. Given $0 < \eta_t \leq \frac{1}{2\gamma L}$, we have*

$$\Phi(x_{t+1}) \leq \Phi(x_t) + \gamma L_{12}\eta_t\|y^*(x_t) - y_t\|^2 + \gamma\eta_t\|grad_x f(x_t, y_t) - v_t\|^2 - \frac{\gamma\eta_t}{2}\|grad\Phi(x_t)\|^2$$

$$- \frac{\gamma\eta_t}{4}\|v_t\|^2.$$  (29)

*Proof.* According to Assumption 2, i.e., the function $\Phi(x)$ is retraction $L$-smooth, we have

$$\Phi(x_{t+1}) \leq \Phi(x_t) - \gamma\eta_t\langle \text{grad}\Phi(x_t), v_t\rangle + \frac{\gamma^2\eta_t^2 L}{2}\|v_t\|^2$$  (30)

$$= \Phi(x_t) + \frac{\gamma\eta_t}{2}\|\text{grad}\Phi(x_t) - v_t\|^2 - \frac{\gamma\eta_t}{2}\|\text{grad}\Phi(x_t)\|^2 + (\frac{\gamma^2\eta_t^2 L}{2} - \frac{\gamma\eta_t}{2})\|v_t\|^2$$

$$= \Phi(x_t) + \frac{\gamma\eta_t}{2}\|\text{grad}\Phi(x_t) - \text{grad}_x f(x_t, y_t) + \text{grad}_x f(x_t, y_t) - v_t\|^2 - \frac{\gamma\eta_t}{2}\|\text{grad}\Phi(x_t)\|^2$$

$$+ (\frac{\gamma^2\eta_t^2 L}{2} - \frac{\gamma\eta_t}{2})\|v_t\|^2$$

$$\leq \Phi(x_t) + \gamma\eta_t\|\text{grad}\Phi(x_t) - \text{grad}_x f(x_t, y_t)\|^2 + \gamma\eta_t\|\text{grad}_x f(x_t, y_t) - v_t\|^2 - \frac{\gamma\eta_t}{2}\|\text{grad}\Phi(x_t)\|^2$$

$$+ (\frac{L\gamma^2\eta_t^2}{2} - \frac{\gamma\eta_t}{2})\|v_t\|^2$$

$$\leq \Phi(x_t) + \gamma\eta_t\|\text{grad}\Phi(x_t) - \text{grad}_x f(x_t, y_t)\|^2 + \gamma\eta_t\|\text{grad}_x f(x_t, y_t) - v_t\|^2 - \frac{\gamma\eta_t}{2}\|\text{grad}\Phi(x_t)\|^2$$

$$- \frac{\gamma\eta_t}{4}\|v_t\|^2,$$

where the last inequality is due to $0 < \eta_t \leq \frac{1}{2\gamma L}$.

Consider an upper bound of $\|\text{grad}\Phi(x_t) - \text{grad}_x f(x_t, y_t)\|^2$, we have

$$\|\text{grad}\Phi(x_t) - \text{grad}_x f(x_t, y_t)\|^2 = \|\text{grad}_x f(x_t, y^*(x_t)) - \text{grad}_x f(x_t, y_t)\|^2$$

$$\leq L_{12}\|y^*(x_t) - y_t\|^2.$$  (31)

Then we have

$$\Phi(x_{t+1}) \leq \Phi(x_t) + \gamma\eta_t L_{12}\|y^*(x_t) - y_t\|^2 + \gamma\eta_t\|\text{grad}_x f(x_t, y_t) - v_t\|^2 - \frac{\gamma\eta_t}{2}\|\text{grad}\Phi(x_t)\|^2$$

$$- \frac{\gamma\eta_t}{4}\|v_t\|^2.$$  (32)

$\square$

**Lemma 6.** *Suppose the sequence $\{x_t, y_t\}_{t=1}^T$ is generated from Algorithm 1 or 2. Under the above assumptions, and set $0 < \eta_t \leq 1$ and $0 < \lambda \leq \frac{1}{6L}$, we have*

$$\|y_{t+1} - y^*(x_{t+1})\|^2 \leq (1 - \frac{\eta_t\mu\lambda}{4})\|y_t - y^*(x_t)\|^2 - \frac{3\eta_t}{4}\|\tilde{y}_{t+1} - y_t\|^2$$

$$+ \frac{25\eta_t\lambda}{6\mu}\|\nabla_y f(x_t, y_t) - w_t\|^2 + \frac{25\gamma^2\kappa^2\eta_t}{6\mu\lambda}\|v_t\|^2,$$  (33)

*where $\kappa = L_{21}/\mu$.*

*Proof.* According to the assumption 3, i.e., the function $f(x, y)$ is $\mu$-strongly concave w.r.t $y$, we have

$$
f(x_t, y) \leq f(x_t, y_t) + \langle \nabla_y f(x_t, y_t), y - y_t \rangle - \frac{\mu}{2} \|y - y_t\|^2
$$
$$
= f(x_t, y_t) + \langle w_t, y - \tilde{y}_{t+1} \rangle + \langle \nabla_y f(x_t, y_t) - w_t, y - \tilde{y}_{t+1} \rangle
$$
$$
+ \langle \nabla_y f(x_t, y_t), \tilde{y}_{t+1} - y_t \rangle - \frac{\mu}{2} \|y - y_t\|^2. \tag{34}
$$

According to the assumption 1, i.e., the function $f(x, y)$ is $L_{22}$-smooth w.r.t $y$, and $\tilde{L} \geq L_{22}$, we have

$$
f(x_t, \tilde{y}_{t+1}) - f(x_t, y_t) - \langle \nabla_y f(x_t, y_t), \tilde{y}_{t+1} - y_t \rangle \geq -\frac{L_{22}}{2} \|\tilde{y}_{t+1} - y_t\|^2
$$
$$
\geq -\frac{\tilde{L}}{2} \|\tilde{y}_{t+1} - y_t\|^2. \tag{35}
$$

Combining the inequalities (34) with (35), we have

$$
f(x_t, y) \leq f(x_t, \tilde{y}_{t+1}) + \langle w_t, y - \tilde{y}_{t+1} \rangle + \langle \nabla_y f(x_t, y_t) - w_t, y - \tilde{y}_{t+1} \rangle
$$
$$
- \frac{\mu}{2} \|y - y_t\|^2 + \frac{\tilde{L}}{2} \|\tilde{y}_{t+1} - y_t\|^2. \tag{36}
$$

According to the step 6 of Algorithm 1 or 2, we have $\tilde{y}_{t+1} = \mathcal{P}_{\mathcal{Y}}(y_t + \lambda w_t) = \arg\min_{y \in \mathcal{Y}} \frac{1}{2} \|y - y_t - \lambda w_t\|^2$. Since $\mathcal{Y}$ is a convex set and the function $\frac{1}{2} \|y - y_t - \lambda w_t\|^2$ is convex, according to Lemma 2, we have

$$
\langle \tilde{y}_{t+1} - y_t - \lambda w_t, y - \tilde{y}_{t+1} \rangle \geq 0, \ y \in \mathcal{Y}. \tag{37}
$$

Then we obtain

$$
\langle w_t, y - \tilde{y}_{t+1} \rangle \leq \frac{1}{\lambda} \langle \tilde{y}_{t+1} - y_t, y - \tilde{y}_{t+1} \rangle
$$
$$
= \frac{1}{\lambda} \langle \tilde{y}_{t+1} - y_t, y_t - \tilde{y}_{t+1} \rangle + \frac{1}{\lambda} \langle \tilde{y}_{t+1} - y_t, y - y_t \rangle
$$
$$
= -\frac{1}{\lambda} \|\tilde{y}_{t+1} - y_t\|^2 + \frac{1}{\lambda} \langle \tilde{y}_{t+1} - y_t, y - y_t \rangle. \tag{38}
$$

Combining the inequalities (36) with (38), we have

$$
f(x_t, y) \leq f(x_t, \tilde{y}_{t+1}) + \frac{1}{\lambda} \langle \tilde{y}_{t+1} - y_t, y - y_t \rangle + \langle \nabla_y f(x_t, y_t) - w_t, y - \tilde{y}_{t+1} \rangle
$$
$$
- \frac{1}{\lambda} \|\tilde{y}_{t+1} - y_t\|^2 - \frac{\mu}{2} \|y - y_t\|^2 + \frac{\tilde{L}}{2} \|\tilde{y}_{t+1} - y_t\|^2. \tag{39}
$$

Let $y = y^*(x_t)$ and we obtain

$$
f(x_t, y^*(x_t)) \leq f(x_t, \tilde{y}_{t+1}) + \frac{1}{\lambda} \langle \tilde{y}_{t+1} - y_t, y^*(x_t) - y_t \rangle + \langle \nabla_y f(x_t, y_t) - w_t, y^*(x_t) - \tilde{y}_{t+1} \rangle
$$
$$
- \frac{1}{\lambda} \|\tilde{y}_{t+1} - y_t\|^2 - \frac{\mu}{2} \|y^*(x_t) - y_t\|^2 + \frac{\tilde{L}}{2} \|\tilde{y}_{t+1} - y_t\|^2. \tag{40}
$$

Due to the concavity of $f(\cdot, y)$ and $y^*(x_t) = \arg\max_{y \in \mathcal{Y}} f(x_t, y)$, we have $f(x_t, y^*(x_t)) \geq f(x_t, \tilde{y}_{t+1})$. Thus, we obtain

$$
0 \leq \frac{1}{\lambda} \langle \tilde{y}_{t+1} - y_t, y^*(x_t) - y_t \rangle + \langle \nabla_y f(x_t, y_t) - w_t, y^*(x_t) - \tilde{y}_{t+1} \rangle
$$
$$
- (\frac{1}{\lambda} - \frac{\tilde{L}}{2}) \|\tilde{y}_{t+1} - y_t\|^2 - \frac{\mu}{2} \|y^*(x_t) - y_t\|^2. \tag{41}
$$

By $y_{t+1} = y_t + \eta_t(\tilde{y}_{t+1} - y_t)$, we have

$$
\|y_{t+1} - y^*(x_t)\|^2 = \|y_t + \eta_t(\tilde{y}_{t+1} - y_t) - y^*(x_t)\|^2
$$
$$
= \|y_t - y^*(x_t)\|^2 + 2\eta_t \langle \tilde{y}_{t+1} - y_t, y_t - y^*(x_t) \rangle + \eta_t^2 \|\tilde{y}_{t+1} - y_t\|^2. \tag{42}
$$

Then we obtain

$$\langle \tilde{y}_{t+1} - y_t, y^*(x_t) - y_t \rangle \le \frac{1}{2\eta_t} \|y_t - y^*(x_t)\|^2 + \frac{\eta_t}{2} \|\tilde{y}_{t+1} - y_t\|^2 - \frac{1}{2\eta_t} \|y_{t+1} - y^*(x_t)\|^2. \quad (43)$$

Consider the upper bound of the term $\langle \nabla_y f(x_t, y_t) - w_t, y^*(x_t) - \tilde{y}_{t+1} \rangle$, we have

$$
\begin{aligned}
&\langle \nabla_y f(x_t, y_t) - w_t, y^*(x_t) - \tilde{y}_{t+1} \rangle \\
&= \langle \nabla_y f(x_t, y_t) - w_t, y^*(x_t) - y_t \rangle + \langle \nabla_y f(x_t, y_t) - w_t, y_t - \tilde{y}_{t+1} \rangle \\
&\le \frac{1}{\mu} \|\nabla_y f(x_t, y_t) - w_t\|^2 + \frac{\mu}{4} \|y^*(x_t) - y_t\|^2 + \frac{1}{\mu} \|\nabla_y f(x_t, y_t) - w_t\|^2 + \frac{\mu}{4} \|y_t - \tilde{y}_{t+1}\|^2 \\
&= \frac{2}{\mu} \|\nabla_y f(x_t, y_t) - w_t\|^2 + \frac{\mu}{4} \|y^*(x_t) - y_t\|^2 + \frac{\mu}{4} \|y_t - \tilde{y}_{t+1}\|^2. \quad (44)
\end{aligned}
$$

By plugging the inequalities (41), (43) to (44), we have

$$
\begin{aligned}
\frac{1}{2\eta_t\lambda} \|y_{t+1} - y^*(x_t)\|^2 &\le (\frac{1}{2\eta_t\lambda} - \frac{\mu}{4}) \|y_t - y^*(x_t)\|^2 + (\frac{\eta_t}{2\lambda} + \frac{\mu}{4} + \frac{\tilde{L}}{2} - \frac{1}{\lambda}) \|\tilde{y}_{t+1} - y_t\|^2 \\
&\quad + \frac{2}{\mu} \|\nabla_y f(x_t, y_t) - w_t\|^2 \\
&\le (\frac{1}{2\eta_t\lambda} - \frac{\mu}{4}) \|y_t - y^*(x_t)\|^2 + (\frac{3\tilde{L}}{4} - \frac{1}{2\lambda}) \|\tilde{y}_{t+1} - y_t\|^2 + \frac{2}{\mu} \|\nabla_y f(x_t, y_t) - w_t\|^2 \\
&= (\frac{1}{2\eta_t\lambda} - \frac{\mu}{4}) \|y_t - y^*(x_t)\|^2 - (\frac{3}{8\lambda} + \frac{1}{8\lambda} - \frac{3\tilde{L}}{4}) \|\tilde{y}_{t+1} - y_t\|^2 \\
&\quad + \frac{2}{\mu} \|\nabla_y f(x_t, y_t) - w_t\|^2 \\
&\le (\frac{1}{2\eta_t\lambda} - \frac{\mu}{4}) \|y_t - y^*(x_t)\|^2 - \frac{3}{8\lambda} \|\tilde{y}_{t+1} - y_t\|^2 + \frac{2}{\mu} \|\nabla_y f(x_t, y_t) - w_t\|^2, \quad (45)
\end{aligned}
$$

where the second inequality holds by $\tilde{L} \ge L_{22} \ge \mu$ and $0 < \eta_t \le 1$, and the last inequality is due to $0 < \lambda \le \frac{1}{6\tilde{L}}$. It implies that

$$\|y_{t+1} - y^*(x_t)\|^2 \le (1 - \frac{\eta_t\mu\lambda}{2}) \|y_t - y^*(x_t)\|^2 - \frac{3\eta_t}{4} \|\tilde{y}_{t+1} - y_t\|^2 + \frac{4\eta_t\lambda}{\mu} \|\nabla_y f(x_t, y_t) - w_t\|^2. \quad (46)$$

Next, we decompose the term $\|y_{t+1} - y^*(x_{t+1})\|^2$ as follows:

$$
\begin{aligned}
\|y_{t+1} - y^*(x_{t+1})\|^2 &= \|y_{t+1} - y^*(x_t) + y^*(x_t) - y^*(x_{t+1})\|^2 \\
&= \|y_{t+1} - y^*(x_t)\|^2 + 2\langle y_{t+1} - y^*(x_t), y^*(x_t) - y^*(x_{t+1}) \rangle + \|y^*(x_t) - y^*(x_{t+1})\|^2 \\
&\le (1 + \frac{\eta_t\mu\lambda}{4}) \|y_{t+1} - y^*(x_t)\|^2 + (1 + \frac{4}{\eta_t\mu\lambda}) \|y^*(x_t) - y^*(x_{t+1})\|^2 \\
&\le (1 + \frac{\eta_t\mu\lambda}{4}) \|y_{t+1} - y^*(x_t)\|^2 + (1 + \frac{4}{\eta_t\mu\lambda}) \eta_t^2 \gamma^2 \kappa^2 \|v_t\|^2, \quad (47)
\end{aligned}
$$

where the first inequality holds by the Cauchy-Schwarz inequality and Young's inequality, and the last equality is due to Lemma 4.

By combining the above inequalities (46) and (47), we have

$$
\begin{aligned}
\|y_{t+1} - y^*(x_{t+1})\|^2 &\le (1 + \frac{\eta_t\mu\lambda}{4})(1 - \frac{\eta_t\mu\lambda}{2}) \|y_t - y^*(x_t)\|^2 - (1 + \frac{\eta_t\mu\lambda}{4}) \frac{3\eta_t}{4} \|\tilde{y}_{t+1} - y_t\|^2 \\
&\quad + (1 + \frac{\eta_t\mu\lambda}{4}) \frac{4\eta_t\lambda}{\mu} \|\nabla_y f(x_t, y_t) - w_t\|^2 + (1 + \frac{4}{\eta_t\mu\lambda}) \eta_t^2 \gamma^2 \kappa^2 \|v_t\|^2. \quad (48)
\end{aligned}
$$

Since $0 < \eta_t \leq 1$, $0 < \lambda \leq \frac{1}{6\tilde{L}}$ and $\tilde{L} \geq L_{22} \geq \mu$, we have $\lambda \leq \frac{1}{6\tilde{L}} \leq \frac{1}{6\mu}$ and $\eta_t \leq 1 \leq \frac{1}{6\mu\lambda}$. Then we obtain

$$(1 + \frac{\eta_t\mu\lambda}{4})(1 - \frac{\eta_t\mu\lambda}{2}) = 1 - \frac{\eta_t\mu\lambda}{2} + \frac{\eta_t\mu\lambda}{4} - \frac{\eta_t^2\mu^2\lambda^2}{8} \leq 1 - \frac{\eta_t\mu\lambda}{4},$$

$$-(1 + \frac{\eta_t\mu\lambda}{4})\frac{3\eta_t}{4} \leq -\frac{3\eta_t}{4},$$

$$(1 + \frac{\eta_t\mu\lambda}{4})\frac{4\eta_t\lambda}{\mu} \leq (1 + \frac{1}{24})\frac{4\eta_t\lambda}{\mu} = \frac{25\eta_t\lambda}{6\mu},$$

$$(1 + \frac{4}{\eta_t\mu\lambda})\gamma^2\kappa^2\eta_t^2 = \gamma^2\kappa^2\eta_t^2 + \frac{4\gamma^2\kappa^2\eta_t}{\mu\lambda} \leq \frac{\gamma^2\kappa^2\eta_t}{6\mu\lambda} + \frac{4\gamma^2\kappa^2\eta_t}{\mu\lambda} = \frac{25\gamma^2\kappa^2\eta_t}{6\mu\lambda}. \tag{49}$$

Thus we have

$$\|y_{t+1} - y^*(x_{t+1})\|^2 \leq (1 - \frac{\eta_t\mu\lambda}{4})\|y_t - y^*(x_t)\|^2 - \frac{3\eta_t}{4}\|\tilde{y}_{t+1} - y_t\|^2$$

$$+ \frac{25\eta_t\lambda}{6\mu}\|\nabla_y f(x_t, y_t) - w_t\|^2 + \frac{25\gamma^2\kappa^2\eta_t}{6\mu\lambda}\|v_t\|^2. \tag{50}$$

$\square$

### A.1 CONVERGENCE ANALYSIS OF RGDA AND RSGDA ALGORITHMS

In the subsection, we study the convergence properties of deterministic RGDA and stochastic RS-GDA algorithms, respectively. For notational simplicity, let $\tilde{L} = \max(1, L_{11}, L_{12}, L_{21}, L_{22})$.

**Theorem 4.** *Suppose the sequence $\{x_t, y_t\}_{t=1}^T$ is generated from Algorithm 1 by using deterministic gradients. Given $\eta = \eta_t$ for all $t \geq 1$, $0 < \eta \leq \min(1, \frac{1}{2\gamma L})$, $0 < \lambda \leq \frac{1}{6\tilde{L}}$ and $0 < \gamma \leq \frac{\mu\lambda}{10\tilde{L}\kappa}$, we have*

$$\frac{1}{T}\sum_{t=1}^T \left[\tilde{L}\|y_t - y^*(x_t)\| + \|grad\Phi(x_t)\|\right] \leq \frac{2\sqrt{\Phi(x_1) - \Phi^*}}{\sqrt{\gamma\eta T}}. \tag{51}$$

*Proof.* According to Lemma 6, we have

$$\|y_{t+1} - y^*(x_{t+1})\|^2 \leq (1 - \frac{\eta_t\mu\lambda}{4})\|y_t - y^*(x_t)\|^2 - \frac{3\eta_t}{4}\|\tilde{y}_{t+1} - y_t\|^2 + \frac{25\eta_t\lambda}{6\mu}\|\nabla_y f(x_t, y_t) - w_t\|^2$$

$$+ \frac{25\gamma^2\kappa^2\eta_t}{6\mu\lambda}\|v_t\|^2. \tag{52}$$

We first define a *Lyapunov* function $\Lambda_t$, for any $t \geq 1$

$$\Lambda_t = \Phi(x_t) + \frac{6\gamma\tilde{L}^2}{\lambda\mu}\|y_t - y^*(x_t)\|^2. \tag{53}$$

According to Lemma 5, we have

$$\Lambda_{t+1} - \Lambda_t = \Phi(x_{t+1}) - \Phi(x_t) + \frac{6\gamma\tilde{L}^2}{\lambda\mu}\left(\|y_{t+1} - y^*(x_{t+1})\|^2 - \|y_t - y^*(x_t)\|^2\right)$$

$$\leq \gamma\eta_t L_{12}\|y_t - y^*(x_t)\|^2 + \gamma\eta_t\|grad_x f(x_t, y_t) - v_t\|^2 - \frac{\gamma\eta_t}{2}\|grad\Phi(x_t)\|^2 - \frac{\gamma\eta_t}{4}\|v_t\|^2$$

$$+ \frac{6\gamma\tilde{L}^2}{\lambda\mu}\left(-\frac{\mu\lambda\eta_t}{4}\|y_t - y^*(x_t)\|^2 - \frac{3\eta_t}{4}\|\tilde{y}_{t+1} - y_t\|^2 + \frac{25\lambda\eta_t}{6\mu}\|\nabla_y f(x_t, y_t) - w_t\|^2\right.$$

$$\left. + \frac{25\gamma^2\kappa^2\eta_t}{6\mu\lambda}\|v_t\|^2\right)$$

$$\leq -\frac{\tilde{L}^2\gamma\eta_t}{2}\|y_t - y^*(x_t)\|^2 - \frac{\gamma\eta_t}{2}\|grad\Phi(x_t)\|^2 - \frac{9\gamma\tilde{L}^2\eta_t}{2\lambda\mu}\|\tilde{y}_{t+1} - y_t\|^2$$

$$- (\frac{1}{4} - \frac{25\kappa^2\tilde{L}^2\gamma^2}{\mu^2\lambda^2})\gamma\eta_t\|v_t\|^2$$

$$\leq -\frac{\tilde{L}^2\gamma\eta_t}{2}\|y_t - y^*(x_t)\|^2 - \frac{\gamma\eta_t}{2}\|grad\Phi(x_t)\|^2, \tag{54}$$

where the first inequality holds by the inequality (52); the second last inequality is due to $\tilde{L} = \max(1, L_{11}, L_{12}, L_{21}, L_{22})$ and $v_t = \text{grad}_x f(x_t, y_t)$, $w_t = \nabla_y f(x_t, y_t)$, and the last inequality is due to $0 < \gamma \leq \frac{\mu\lambda}{10\tilde{L}\kappa}$. Thus, we obtain

$$\frac{\tilde{L}^2\gamma\eta_t}{2}\|y_t - y^*(x_t)\|^2 + \frac{\gamma\eta_t}{2}\|\text{grad}\Phi(x_t)\|^2 \leq \Lambda_t - \Lambda_{t+1}. \tag{55}$$

Since the initial solution satisfies $y_1 = y^*(x_1) = \arg\max_{y \in \mathcal{Y}} f(x_1, y)$, we have

$$\Lambda_1 = \Phi(x_1) + \frac{6\gamma\tilde{L}^2}{\lambda\mu}\|y_1 - y^*(x_1)\|^2 = \Phi(x_1). \tag{56}$$

Taking average over $t = 1, 2, \cdots, T$ on both sides of the inequality (55), we have

$$\frac{1}{T}\sum_{t=1}^{T}\left[\frac{\tilde{L}^2\eta_t}{2}\|y_t - y^*(x_t)\|^2 + \frac{\eta_t}{2}\|\text{grad}\Phi(x_t)\|^2\right] \leq \frac{\Lambda_1 - \Lambda_{T+1}}{\gamma T} \leq \frac{\Phi(x_1) - \Phi^*}{\gamma T}, \tag{57}$$

where the last equality is due to the above equality (56) and Assumption 4. Let $\eta = \eta_1 = \cdots = \eta_T$, we have

$$\frac{1}{T}\sum_{t=1}^{T}\left[\tilde{L}^2\|y_t - y^*(x_t)\|^2 + \|\text{grad}\Phi(x_t)\|^2\right] \leq \frac{2(\Phi(x_1) - \Phi^*)}{\gamma\eta T}. \tag{58}$$

According to Jensen's inequality, we have

$$\frac{1}{T}\sum_{t=1}^{T}\left[\tilde{L}\|y_t - y^*(x_t)\| + \|\text{grad}\Phi(x_t)\|\right] \leq \left(\frac{2}{T}\sum_{t=1}^{T}\left[\tilde{L}^2\|y_t - y^*(x_t)\|^2 + \|\text{grad}\Phi(x_t)\|^2\right]\right)^{1/2}$$

$$\leq \left(\frac{4(\Phi(x_1) - \Phi^*)}{\gamma\eta T}\right)^{1/2} = \frac{2\sqrt{\Phi(x_1) - \Phi^*}}{\sqrt{\gamma\eta T}}. \tag{59}$$

$\square$

**Theorem 5.** *Suppose the sequence* $\{x_t, y_t\}_{t=1}^{T}$ *is generated from Algorithm 1 by using stochastic gradients. Given* $\eta = \eta_t$ *for all* $t \geq 1$, $0 < \eta \leq \min(1, \frac{1}{2\gamma L})$, $0 < \lambda \leq \frac{1}{6\tilde{L}}$ *and* $0 < \gamma \leq \frac{\mu\lambda}{10\tilde{L}\kappa}$, *we have*

$$\frac{1}{T}\sum_{t=1}^{T}\mathbb{E}\left[\tilde{L}\|y_t - y^*(x_t)\| + \|grad\Phi(x_t)\|\right] \leq \frac{2\sqrt{\Phi(x_1) - \Phi^*}}{\sqrt{\gamma\eta T}} + \frac{\sqrt{2}\sigma}{\sqrt{B}} + \frac{5\sqrt{2}\tilde{L}\sigma}{\sqrt{B}\mu}. \tag{60}$$

*Proof.* According to Lemma 6, we have

$$\|y_{t+1} - y^*(x_{t+1})\|^2 \leq (1 - \frac{\eta_t\mu\lambda}{4})\|y_t - y^*(x_t)\|^2 - \frac{3\eta_t}{4}\|\tilde{y}_{t+1} - y_t\|^2 + \frac{25\eta_t\lambda}{6\mu}\|\nabla_y f(x_t, y_t) - w_t\|^2$$

$$+ \frac{25\gamma^2\kappa^2\eta_t}{6\mu\lambda}\|v_t\|^2. \tag{61}$$

We first define a *Lyapunov* function $\Theta_t$, for any $t \geq 1$

$$\Theta_t = \mathbb{E}\left[\Phi(x_t) + \frac{6\gamma\tilde{L}^2}{\lambda\mu}\|y_t - y^*(x_t)\|^2\right]. \tag{62}$$

By Assumption 5, we have

$$\mathbb{E}\|\text{grad}_x f(x_t, y_t) - v_t\|^2 = \mathbb{E}\|\text{grad}_x f(x_t, y_t) - \frac{1}{B}\sum_{i=1}^{B}\text{grad}_x f(x_t, y_t; \xi_t^i)\|^2 \leq \frac{\sigma^2}{B}, \tag{63}$$

$$\mathbb{E}\|\nabla_y f(x_t, y_t) - w_t\|^2 = \mathbb{E}\|\nabla_y f(x_t, y_t) - \frac{1}{B}\sum_{i=1}^{B}\nabla_y f(x_t, y_t; \xi_t^i)\|^2 \leq \frac{\sigma^2}{B}. \tag{64}$$

According to Lemma 5, we have

$$
\begin{aligned}
\Theta_{t+1} - \Theta_t &= \mathbb{E}[\Phi(x_{t+1})] - \mathbb{E}[\Phi(x_t)] + \frac{6\gamma\tilde{L}^2}{\lambda\mu}\big(\mathbb{E}\|y_{t+1} - y^*(x_{t+1})\|^2 - \mathbb{E}\|y_t - y^*(x_t)\|^2\big) \\
&\leq \gamma\eta_t L_{12}\mathbb{E}\|y_t - y^*(x_t)\|^2 + \gamma\eta_t\mathbb{E}\|\mathrm{grad}_x f(x_t, y_t) - v_t\|^2 - \frac{\gamma\eta_t}{2}\mathbb{E}\|\mathrm{grad}\Phi(x_t)\|^2 - \frac{\gamma\eta_t}{4}\|v_t\|^2 \\
&\quad + \frac{6\gamma\tilde{L}^2}{\lambda\mu}\Big(-\frac{\mu\lambda\eta_t}{4}\mathbb{E}\|y_t - y^*(x_t)\|^2 - \frac{3\eta_t}{4}\mathbb{E}\|\tilde{y}_{t+1} - y_t\|^2 + \frac{25\lambda\eta_t}{6\mu}\mathbb{E}\|\nabla_y f(x_t, y_t) - w_t\|^2 \\
&\quad + \frac{25\gamma^2\kappa^2\eta_t}{6\mu\lambda}\|v_t\|^2\Big) \\
&\leq -\frac{\tilde{L}^2\gamma\eta_t}{2}\mathbb{E}\|y_t - y^*(x_t)\|^2 - \frac{\gamma\eta_t}{2}\mathbb{E}\|\mathrm{grad}\Phi(x_t)\|^2 - \frac{9\gamma\tilde{L}^2\eta_t}{2\lambda\mu}\mathbb{E}\|\tilde{y}_{t+1} - y_t\|^2 \\
&\quad - \big(\frac{1}{4} - \frac{25\kappa^2\tilde{L}^2\gamma^2}{\mu^2\lambda^2}\big)\gamma\eta_t\|v_t\|^2 + \gamma\eta_t\mathbb{E}\|\mathrm{grad}_x f(x_t, y_t) - v_t\|^2 + \frac{25\tilde{L}^2\gamma\eta_t}{\mu^2}\mathbb{E}\|\nabla_y f(x_t, y_t) - w_t\|^2 \\
&\leq -\frac{\tilde{L}^2\gamma\eta_t}{2}\mathbb{E}\|y_t - y^*(x_t)\|^2 - \frac{\gamma\eta_t}{2}\mathbb{E}\|\mathrm{grad}\Phi(x_t)\|^2 + \frac{\gamma\eta_t\sigma^2}{B} + \frac{25\tilde{L}^2\gamma\eta_t\sigma^2}{B\mu^2}, \qquad (65)
\end{aligned}
$$

where the first inequality holds by the inequality (61); the second last inequality is due to $\tilde{L} = \max(1, L_{11}, L_{12}, L_{21}, L_{22})$, and the last inequality is due to $0 < \gamma \leq \frac{\mu\lambda}{10\tilde{L}\kappa}$ and Assumption 5. Thus, we obtain

$$
\frac{\tilde{L}^2\gamma\eta_t}{2}\mathbb{E}\|y_t - y^*(x_t)\|^2 + \frac{\gamma\eta_t}{2}\mathbb{E}\|\mathrm{grad}\Phi(x_t)\|^2 \leq \Theta_t - \Theta_{t+1} + \frac{\gamma\eta_t\sigma^2}{B} + \frac{25\tilde{L}^2\gamma\eta_t\sigma^2}{B\mu^2}. \qquad (66)
$$

Since the initial solution satisfies $y_1 = y^*(x_1) = \arg\max_{y\in\mathcal{Y}} f(x_1, y)$, we have

$$
\Theta_1 = \Phi(x_1) + \frac{6\gamma\tilde{L}^2}{\lambda\mu}\|y_1 - y^*(x_1)\|^2 = \Phi(x_1). \qquad (67)
$$

Taking average over $t = 1, 2, \cdots, T$ on both sides of the inequality (66), we have

$$
\begin{aligned}
\frac{1}{T}\sum_{t=1}^T \mathbb{E}\big[\frac{\tilde{L}^2\eta_t}{2}\|y_t - y^*(x_t)\|^2 + \frac{\eta_t}{2}\|\mathrm{grad}\Phi(x_t)\|^2\big] &\leq \frac{\Theta_t - \Theta_{t+1}}{\gamma T} + \frac{1}{T}\sum_{t=1}^T \frac{\eta_t\sigma^2}{B} + \frac{1}{T}\sum_{t=1}^T \frac{25\tilde{L}^2\eta_t\sigma^2}{B\mu^2} \\
&= \frac{\Phi(x_1) - \Phi^*}{\gamma T} + \frac{1}{T}\sum_{t=1}^T \frac{\eta_t\sigma^2}{B} + \frac{1}{T}\sum_{t=1}^T \frac{25\tilde{L}^2\eta_t\sigma^2}{B\mu^2}, \\
&\qquad\qquad\qquad\qquad\qquad\qquad\qquad\qquad\qquad\qquad\qquad (68)
\end{aligned}
$$

where the last equality is due to the above equality (67). Let $\eta = \eta_1 = \cdots = \eta_T$, we have

$$
\frac{1}{T}\sum_{t=1}^T \mathbb{E}\big[\tilde{L}^2\|y_t - y^*(x_t)\|^2 + \|\mathrm{grad}\Phi(x_t)\|^2\big] \leq \frac{2(\Phi(x_1) - \Phi^*)}{\gamma\eta T} + \frac{\sigma^2}{B} + \frac{25\tilde{L}^2\sigma^2}{B\mu^2}. \qquad (69)
$$

According to Jensen's inequality, we have

$$
\begin{aligned}
\frac{1}{T}\sum_{t=1}^T \mathbb{E}\big[\tilde{L}\|y_t - y^*(x_t)\| + \|\mathrm{grad}\Phi(x_t)\|\big] &\leq \big(\frac{2}{T}\sum_{t=1}^T \mathbb{E}\big[\tilde{L}^2\|y_t - y^*(x_t)\|^2 + \|\mathrm{grad}\Phi(x_t)\|^2\big]^{1/2} \\
&\leq \frac{4(\Phi(x_1) - \Phi^*)}{\gamma\eta T} + \frac{2\sigma^2}{B} + \frac{50\tilde{L}^2\sigma^2}{B\mu^2}\big)^{1/2} \\
&\leq \frac{2\sqrt{\Phi(x_1) - \Phi^*}}{\sqrt{\gamma\eta T}} + \frac{\sqrt{2}\sigma}{\sqrt{B}} + \frac{5\sqrt{2}\tilde{L}\sigma}{\sqrt{B}\mu}, \qquad (70)
\end{aligned}
$$

where the last inequality is due to $(a_1 + a_2 + a_3)^{1/2} \leq a_1^{1/2} + a_2^{1/2} + a_3^{1/2}$ for all $a_1, a_2, a_3 > 0$.

$\square$

### A.2 Convergence Analysis of the MVR-RSGDA Algorithm

In the subsection, we study the convergence properties of the MVR-RSGDA algorithm. For notational simplicity, let $\tilde{L} = \max(L_{11}, L_{12}, L_{21}, L_{22}, 1)$.

**Lemma 7.** *Suppose the stochastic gradients $v_t$ and $w_t$ is generated from Algorithm 2, given $0 < \alpha_{t+1} \leq 1$ and $0 < \beta_{t+1} \leq 1$, we have*

$$\mathbb{E}\|grad_x f(x_{t+1}, y_{t+1}) - v_{t+1}\|^2 \leq (1 - \alpha_{t+1})^2 \mathbb{E}\|grad_x f(x_t, y_t) - v_t\|^2 + 4(1 - \alpha_{t+1})^2 L_{11}^2 \gamma^2 \eta_t^2 \|v_t\|^2$$
$$+ 4(1 - \alpha_{t+1})^2 L_{12}^2 \eta_t^2 \|\tilde{y}_{t+1} - y_t\|^2 + \frac{2\alpha_{t+1}^2 \sigma^2}{B}. \tag{71}$$

$$\mathbb{E}\|\nabla_y f(x_{t+1}, y_{t+1}) - w_{t+1}\|^2 \leq (1 - \beta_{t+1})^2 \mathbb{E}\|\nabla_y f(x_t, y_t) - w_t\|^2 + 4(1 - \beta_{t+1})^2 L_{21}^2 \gamma^2 \eta_t^2 \|v_t\|^2$$
$$+ 4(1 - \beta_{t+1})^2 L_{22}^2 \eta_t^2 \|\tilde{y}_{t+1} - y_t\|^2 + \frac{2\beta_{t+1}^2 \sigma^2}{B}. \tag{72}$$

*Proof.* We first prove the inequality (71). According to the definition of $v_t$ in Algorithm 2, we have

$$v_{t+1} - \mathcal{T}_{x_t}^{x_{t+1}} v_t = -\alpha_{t+1} \mathcal{T}_{x_t}^{x_{t+1}} v_t + (1 - \alpha_{t+1})\big(grad_x f_{\mathcal{B}_{t+1}}(x_{t+1}, y_{t+1}) - \mathcal{T}_{x_t}^{x_{t+1}} grad_x f_{\mathcal{B}_{t+1}}(x_t, y_t)\big)$$
$$+ \alpha_{t+1} grad_x f_{\mathcal{B}_{t+1}}(x_{t+1}, y_{t+1}). \tag{73}$$

Then we have

$$\mathbb{E}\|grad_x f(x_{t+1}, y_{t+1}) - v_{t+1}\|^2 \tag{74}$$
$$= \mathbb{E}\|grad_x f(x_{t+1}, y_{t+1}) - \mathcal{T}_{x_t}^{x_{t+1}} v_t - (v_{t+1} - \mathcal{T}_{x_t}^{x_{t+1}} v_t)\|^2$$
$$= \mathbb{E}\|grad_x f(x_{t+1}, y_{t+1}) - \mathcal{T}_{x_t}^{x_{t+1}} v_t + \alpha_{t+1} \mathcal{T}_{x_t}^{x_{t+1}} v_t - \alpha_{t+1} grad_x f_{\mathcal{B}_{t+1}}(x_{t+1}, y_{t+1})$$
$$\quad - (1 - \alpha_{t+1})\big(grad_x f_{\mathcal{B}_{t+1}}(x_{t+1}, y_{t+1}) - \mathcal{T}_{x_t}^{x_{t+1}} grad_x f_{\mathcal{B}_{t+1}}(x_t, y_t)\big)\|^2$$
$$= \mathbb{E}\|(1 - \alpha_{t+1})\mathcal{T}_{x_t}^{x_{t+1}}\big(grad_x f(x_t, y_t) - v_t\big) + (1 - \alpha_{t+1})\big(grad_x f(x_{t+1}, y_{t+1}) - \mathcal{T}_{x_t}^{x_{t+1}} grad_x f(x_t, y_t)$$
$$\quad - grad_x f_{\mathcal{B}_{t+1}}(x_{t+1}, y_{t+1}) + \mathcal{T}_{x_t}^{x_{t+1}} grad_x f_{\mathcal{B}_{t+1}}(x_t, y_t)\big)$$
$$\quad + \alpha_{t+1}\big(grad_x f(x_{t+1}, y_{t+1}) - grad_x f_{\mathcal{B}_{t+1}}(x_{t+1}, y_{t+1})\big)\|^2$$
$$= (1 - \alpha_{t+1})^2 \mathbb{E}\|grad_x f(x_t, y_t) - v_t\|^2 + \alpha_{t+1}^2 \mathbb{E}\|grad_x f(x_{t+1}, y_{t+1}) - grad_x f_{\mathcal{B}_{t+1}}(x_{t+1}, y_{t+1})\|^2$$
$$\quad + (1 - \alpha_{t+1})^2 \mathbb{E}\|grad_x f(x_{t+1}, y_{t+1}) - \mathcal{T}_{x_t}^{x_{t+1}} grad_x f(x_t, y_t) - grad_x f_{\mathcal{B}_{t+1}}(x_{t+1}, y_{t+1})$$
$$\quad + \mathcal{T}_{x_t}^{x_{t+1}} grad_x f_{\mathcal{B}_{t+1}}(x_t, y_t)\|^2 + 2\alpha_{t+1}(1 - \alpha_{t+1})\big\langle grad_x f(x_{t+1}, y_{t+1}) - \mathcal{T}_{x_t}^{x_{t+1}} grad_x f(x_t, y_t)$$
$$\quad - grad_x f_{\mathcal{B}_{t+1}}(x_{t+1}, y_{t+1}) + \mathcal{T}_{x_t}^{x_{t+1}} grad_x f_{\mathcal{B}_{t+1}}(x_t, y_t), grad_x f(x_{t+1}, y_{t+1}) - grad_x f_{\mathcal{B}_{t+1}}(x_{t+1}, y_{t+1})\big\rangle$$
$$\leq (1 - \alpha_{t+1})^2 \mathbb{E}\|grad_x f(x_t, y_t) - v_t\|^2 + 2\alpha_{t+1}^2 \mathbb{E}\|grad_x f(x_{t+1}, y_{t+1}) - grad_x f_{\mathcal{B}_{t+1}}(x_{t+1}, y_{t+1})\|^2$$
$$\quad + 2(1 - \alpha_{t+1})^2 \mathbb{E}\|grad_x f(x_{t+1}, y_{t+1}) - \mathcal{T}_{x_t}^{x_{t+1}} grad_x f(x_t, y_t) - grad_x f_{\mathcal{B}_{t+1}}(x_{t+1}, y_{t+1})$$
$$\quad + \mathcal{T}_{x_t}^{x_{t+1}} grad_x f_{\mathcal{B}_{t+1}}(x_t, y_t)\|^2$$
$$\leq (1 - \alpha_{t+1})^2 \mathbb{E}\|grad_x f(x_t, y_t) - v_t\|^2 + \frac{2\alpha_{t+1}^2 \sigma^2}{B}$$
$$\quad + 2(1 - \alpha_{t+1})^2 \underbrace{\mathbb{E}\|grad_x f_{\mathcal{B}_{t+1}}(x_{t+1}, y_{t+1}) - \mathcal{T}_{x_t}^{x_{t+1}} grad_x f_{\mathcal{B}_{t+1}}(x_t, y_t)\|^2}_{=T_1},$$

where the fourth equality follows by $\mathbb{E}[grad_x f_{\mathcal{B}_{t+1}}(x_{t+1}, y_{t+1})] = grad_x f(x_{t+1}, y_{t+1})$ and $\mathbb{E}[grad_x f_{\mathcal{B}_{t+1}}(x_{t+1}, y_{t+1}) - grad_x f_{\mathcal{B}_{t+1}}(x_t, y_t)] = grad_x f(x_{t+1}, y_{t+1}) - grad_x f(x_t, y_t)$; the first inequality holds by Young's inequality; the last inequality is due to the equality $\mathbb{E}\|\zeta - \mathbb{E}[\zeta]\|^2 = \mathbb{E}\|\zeta\|^2 - \|\mathbb{E}[\zeta]\|^2$ and Assumption 5.

Next, we consider an upper bound of the above term $T_1$ as follows:

$$T_1 = \mathbb{E}\big\|\mathrm{grad}_x f_{\mathcal{B}_{t+1}}(x_{t+1}, y_{t+1}) - \mathcal{T}_{x_t}^{x_{t+1}}\mathrm{grad}_x f_{\mathcal{B}_{t+1}}(x_t, y_t)\big\|^2 \tag{75}$$

$$= \mathbb{E}\big\|\mathrm{grad}_x f_{\mathcal{B}_{t+1}}(x_{t+1}, y_{t+1}) - \mathcal{T}_{x_t}^{x_{t+1}}\mathrm{grad}_x f(x_t, y_{t+1}; \xi_{t+1}) + \mathcal{T}_{x_t}^{x_{t+1}}\mathrm{grad}_x f(x_t, y_{t+1}; \xi_{t+1})$$

$$- \mathcal{T}_{x_t}^{x_{t+1}}\mathrm{grad}_x f_{\mathcal{B}_{t+1}}(x_t, y_t)\big\|^2$$

$$\leq 2\mathbb{E}\big\|\mathrm{grad}_x f_{\mathcal{B}_{t+1}}(x_{t+1}, y_{t+1}) - \mathcal{T}_{x_t}^{x_{t+1}}\mathrm{grad}_x f(x_t, y_{t+1}; \xi_{t+1})\big\|^2$$

$$+ 2\mathbb{E}\big\|\mathrm{grad}_x f(x_t, y_{t+1}; \xi_{t+1}) - \mathrm{grad}_x f_{\mathcal{B}_{t+1}}(x_t, y_t)\big\|^2$$

$$\leq 2L_{11}^2\gamma^2\eta_t^2\|v_t\|^2 + 2L_{12}^2\|y_{t+1} - y_t\|^2$$

$$= 2L_{11}^2\gamma^2\eta_t^2\|v_t\|^2 + 2L_{12}^2\eta_t^2\|\tilde{y}_{t+1} - y_t\|^2, \tag{76}$$

where the last inequality is due to Assumption 1. Thus, we have

$$\mathbb{E}\|\mathrm{grad}_x f(x_{t+1}, y_{t+1}) - v_{t+1}\|^2 \leq (1 - \alpha_{t+1})^2\mathbb{E}\|\mathrm{grad}_x f(x_t, y_t) - v_t\|^2 + 4(1 - \alpha_{t+1})^2 L_{11}^2\gamma^2\eta_t^2\|v_t\|^2$$

$$+ 4(1 - \alpha_{t+1})^2 L_{12}^2\eta_t^2\|\tilde{y}_{t+1} - y_t\|^2 + \frac{2\alpha_{t+1}^2\sigma^2}{B}. \tag{77}$$

We apply a similar analysis to prove the above inequality (72). We obtain

$$\mathbb{E}\|\nabla_y f(x_{t+1}, y_{t+1}) - w_{t+1}\|^2 \leq (1 - \beta_{t+1})^2\mathbb{E}\|\nabla_y f(x_t, y_t) - w_t\|^2 + 4(1 - \beta_{t+1})^2 L_{21}^2\gamma^2\eta_t^2\|v_t\|^2$$

$$+ 4(1 - \beta_{t+1})^2 L_{22}^2\eta_t^2\|\tilde{y}_{t+1} - y_t\|^2 + \frac{2\beta_{t+1}^2\sigma^2}{B}. \tag{78}$$

$\square$

**Theorem 6.** *Suppose the sequence $\{x_t, y_t\}_{t=1}^T$ is generated from Algorithm 2. Given $y_1 = y^*(x_1)$, $c_1 \geq \frac{2}{3b^3} + 2\lambda\mu$, $c_2 \geq \frac{2}{3b^3} + \frac{50\lambda\tilde{L}^2}{\mu}$, $b > 0$, $m \geq \max\big(2, (\tilde{c}b)^3\big)$, $0 < \gamma \leq \frac{\mu\lambda}{2\kappa\tilde{L}\sqrt{25+4\mu\lambda}}$ and $0 < \lambda \leq \frac{1}{6L}$, we have*

$$\frac{1}{T}\sum_{t=1}^T \mathbb{E}\big[\|grad\Phi(x_t)\| + \tilde{L}\|y_t - y^*(x_t)\|\big] \leq \frac{\sqrt{2M'}m^{1/6}}{T^{1/2}} + \frac{\sqrt{2M'}}{T^{1/3}}, \tag{79}$$

*where $\tilde{c} = \max(2\gamma L, c_1, c_2, 1)$ and $M' = \frac{2(\Phi(x_1)-\Phi^*)}{\gamma b} + \frac{2\sigma^2}{\lambda\mu\eta_0 bB} + \frac{2(c_1^2+c_2^2)\sigma^2 b^2}{\lambda\mu B}\ln(m+T)$.*

*Proof.* Since $\eta_t$ is decreasing and $m \geq b^3$, we have $\eta_t \leq \eta_0 = \frac{b}{m^{1/3}} \leq 1$. Similarly, due to $m \geq (2\gamma Lb)^3$, we have $\eta_t \leq \eta_0 = \frac{b}{m^{1/3}} \leq \frac{1}{2\gamma L}$. Due to $0 < \eta_t \leq 1$ and $m \geq \max\big((c_1 b)^3, (c_2 b)^3\big)$, we have $\alpha_{t+1} = c_1\eta_t^2 \leq c_1\eta_t \leq \frac{c_1 b}{m^{1/3}} \leq 1$ and $\beta_{t+1} = c_2\eta_t^2 \leq c_2\eta_t \leq \frac{c_2 b}{m^{1/3}} \leq 1$. According to Lemma 7, we have

$$\frac{1}{\eta_t}\mathbb{E}\|\mathrm{grad}_x f(x_{t+1}, y_{t+1}) - v_{t+1}\|^2 - \frac{1}{\eta_{t-1}}\mathbb{E}\|\mathrm{grad}_x f(x_t, y_t) - v_t\|^2 \tag{80}$$

$$\leq \big(\frac{(1-\alpha_{t+1})^2}{\eta_t} - \frac{1}{\eta_{t-1}}\big)\mathbb{E}\|\mathrm{grad}_x f(x_t, y_t) - v_t\|^2 + 4(1-\alpha_{t+1})^2 L_{11}^2\gamma^2\eta_t\|v_t\|^2$$

$$+ 4(1-\alpha_{t+1})^2 L_{12}^2\eta_t\|\tilde{y}_{t+1} - y_t\|^2 + \frac{2\alpha_{t+1}^2\sigma^2}{\eta_t B}$$

$$\leq \big(\frac{1-\alpha_{t+1}}{\eta_t} - \frac{1}{\eta_{t-1}}\big)\mathbb{E}\|\mathrm{grad}_x f(x_t, y_t) - v_t\|^2 + 4L_{11}^2\gamma^2\eta_t\|v_t\|^2 + 4L_{12}^2\eta_t\|\tilde{y}_{t+1} - y_t\|^2 + \frac{2\alpha_{t+1}^2\sigma^2}{\eta_t B}$$

$$= \big(\frac{1}{\eta_t} - \frac{1}{\eta_{t-1}} - c_1\eta_t\big)\mathbb{E}\|\mathrm{grad}_x f(x_t, y_t) - v_t\|^2 + 4L_{11}^2\gamma^2\eta_t\|v_t\|^2 + 4L_{12}^2\eta_t\|\tilde{y}_{t+1} - y_t\|^2 + \frac{2\alpha_{t+1}^2\sigma^2}{\eta_t B},$$

where the second inequality is due to $0 < \alpha_{t+1} \leq 1$. By a similar way, we also obtain

$$\frac{1}{\eta_t}\mathbb{E}\|\nabla_y f(x_{t+1}, y_{t+1}) - w_{t+1}\|^2 - \frac{1}{\eta_{t-1}}\mathbb{E}\|\nabla_y f(x_t, y_t) - w_t\|^2 \tag{81}$$

$$\leq \big(\frac{1}{\eta_t} - \frac{1}{\eta_{t-1}} - c_2\eta_t\big)\mathbb{E}\|\nabla_y f(x_t, y_t) - w_t\|^2 + 4L_{21}^2\gamma^2\eta_t\|v_t\|^2 + 4L_{22}^2\eta_t\|\tilde{y}_{t+1} - y_t\|^2 + \frac{2\beta_{t+1}^2\sigma^2}{\eta_t B}.$$

By $\eta_t = \frac{b}{(m+t)^{1/3}}$, we have

$$
\begin{aligned}
\frac{1}{\eta_t} - \frac{1}{\eta_{t-1}} &= \frac{1}{b}\big((m+t)^{\frac{1}{3}} - (m+t-1)^{\frac{1}{3}}\big) \\
&\leq \frac{1}{3b(m+t-1)^{2/3}} \leq \frac{1}{3b\big(m/2+t\big)^{2/3}} \\
&\leq \frac{2^{2/3}}{3b(m+t)^{2/3}} = \frac{2^{2/3}}{3b^3}\frac{b^2}{(m/2+t)^{2/3}} = \frac{2^{2/3}}{3b^3}\eta_t^2 \leq \frac{2}{3b^3}\eta_t,
\end{aligned}
\tag{82}
$$

where the first inequality holds by the concavity of function $f(x) = x^{1/3}$, i.e., $(x+y)^{1/3} \leq x^{1/3} + \frac{y}{3x^{2/3}}$; the second inequality is due to $m \geq 2$, and the last inequality is due to $0 < \eta_t \leq 1$. Let $c_1 \geq \frac{2}{3b^3} + 2\lambda\mu$, we have

$$
\frac{1}{\eta_t}\mathbb{E}\|\mathrm{grad}_x f(x_{t+1}, y_{t+1}) - v_{t+1}\|^2 - \frac{1}{\eta_{t-1}}\mathbb{E}\|\mathrm{grad}_x f(x_t, y_t) - v_t\|^2 \tag{83}
$$
$$
\leq -2\lambda\mu\eta_t\mathbb{E}\|\mathrm{grad}_x f(x_t, y_t) - v_t\|^2 + 4L_{11}^2\gamma^2\eta_t\|v_t\|^2 + 4L_{12}^2\eta_t\|\tilde{y}_{t+1} - y_t\|^2 + \frac{2\alpha_{t+1}^2\sigma^2}{\eta_t B}.
$$

Let $c_2 \geq \frac{2}{3b^3} + \frac{50\lambda\tilde{L}^2}{\mu}$, we have

$$
\frac{1}{\eta_t}\mathbb{E}\|\nabla_y f(x_{t+1}, y_{t+1}) - w_{t+1}\|^2 - \frac{1}{\eta_{t-1}}\mathbb{E}\|\nabla_y f(x_t, y_t) - w_t\|^2 \tag{84}
$$
$$
\leq -\frac{50\lambda\tilde{L}^2}{\mu}\eta_t\mathbb{E}\|\nabla_y f(x_t, y_t) - w_t\|^2 + 4L_{21}^2\gamma^2\eta_t\|v_t\|^2 + 4L_{22}^2\eta_t\|\tilde{y}_{t+1} - y_t\|^2 + \frac{2\beta_{t+1}^2\sigma^2}{\eta_t B}.
$$

According to Lemma 6, we have

$$
\begin{aligned}
\|y_{t+1} - y^*(x_{t+1})\|^2 - \|y_t - y^*(x_t)\|^2 \leq &-\frac{\eta_t\mu\lambda}{4}\|y_t - y^*(x_t)\|^2 - \frac{3\eta_t}{4}\|\tilde{y}_{t+1} - y_t\|^2 \\
&+ \frac{25\lambda\eta_t}{6\mu}\|\nabla_y f(x_t, y_t) - w_t\|^2 + \frac{25\gamma^2\kappa^2\eta_t}{6\mu\lambda}\|v_t\|^2.
\end{aligned}
\tag{85}
$$

Next, we define a *Lyapunov* function $\Omega_t$, for any $t \geq 1$

$$
\begin{aligned}
\Omega_t = &\mathbb{E}\big[\Phi(x_t)\big] + \frac{\gamma}{2\lambda\mu}\Big(\frac{1}{\eta_{t-1}}\mathbb{E}\|\mathrm{grad}_x f(x_t, y_t) - v_t\|^2 + \frac{1}{\eta_{t-1}}\mathbb{E}\|\nabla_y f(x_t, y_t) - w_t\|^2\Big) \\
&+ \frac{6\gamma\tilde{L}^2}{\lambda\mu}\mathbb{E}\|y_t - y^*(x_t)\|^2.
\end{aligned}
\tag{86}
$$

Then we have

$$
\begin{aligned}
\Omega_{t+1} - \Omega_t &= \mathbb{E}[\Phi(x_{t+1})] - \mathbb{E}[\Phi(x_t)] + \frac{6\gamma\tilde{L}^2}{\lambda\mu}\big(\mathbb{E}\|y_{t+1} - y^*(x_{t+1})\|^2 - \mathbb{E}\|y_t - y^*(x_t)\|^2\big) \\
&\quad + \frac{\gamma}{2\lambda\mu}\big(\frac{1}{\eta_t}\mathbb{E}\|\mathrm{grad}_x f(x_{t+1}, y_{t+1}) - v_{t+1}\|^2 - \frac{1}{\eta_{t-1}}\mathbb{E}\|\mathrm{grad}_x f(x_t, y_t) - v_t\|^2 \\
&\quad + \frac{1}{\eta_t}\mathbb{E}\|\nabla_y f(x_{t+1}, y_{t+1}) - w_{t+1}\|^2 - \frac{1}{\eta_{t-1}}\mathbb{E}\|\nabla_y f(x_t, y_t) - w_t\|^2\big) \\
&\leq L_{12}\gamma\eta_t\mathbb{E}\|y_t - y^*(x_t)\|^2 + \gamma\eta_t\mathbb{E}\|\mathrm{grad}_x f(x_t, y_t) - v_t\|^2 - \frac{\gamma\eta_t}{2}\mathbb{E}\|\mathrm{grad}\Phi(x_t)\|^2 - \frac{\gamma\eta_t}{4}\|v_t\|^2 \\
&\quad + \frac{6\gamma\tilde{L}^2}{\lambda\mu}\big(-\frac{\mu\lambda\eta_t}{4}\mathbb{E}\|y_t - y^*(x_t)\|^2 - \frac{3\eta_t}{4}\mathbb{E}\|\tilde{y}_{t+1} - y_t\|^2 + \frac{25\lambda\eta_t}{6\mu}\mathbb{E}\|\nabla_y f(x_t, y_t) - w_t\|^2 + \frac{25\gamma^2\kappa^2\eta_t}{6\mu\lambda}\|v_t\|^2\big) \\
&\quad + \frac{\gamma}{2\lambda\mu}\big(-2\lambda\mu\eta_t\mathbb{E}\|\mathrm{grad}_x f(x_t, y_t) - v_t\|^2 + 4L_{11}^2\gamma^2\eta_t\|v_t\|^2 + 4L_{12}^2\eta_t\mathbb{E}\|\tilde{y}_{t+1} - y_t\|^2 + \frac{2\alpha_{t+1}^2\sigma^2}{\eta_t B} \\
&\quad - \frac{50\lambda\tilde{L}^2}{\mu}\eta_t\mathbb{E}\|\nabla_y f(x_t, y_t) - w_t\|^2 + 4L_{21}^2\gamma^2\eta_t\|v_t\|^2 + 4L_{22}^2\eta_t\mathbb{E}\|\tilde{y}_{t+1} - y_t\|^2 + \frac{2\beta_{t+1}^2\sigma^2}{\eta_t B}\big) \\
&\leq -\frac{\gamma\tilde{L}^2\eta_t}{2}\mathbb{E}\|y_t - y^*(x_t)\|^2 - \frac{\gamma\eta_t}{2}\mathbb{E}\|\mathrm{grad}\Phi(x_t)\|^2 - \frac{\gamma\tilde{L}^2\eta_t}{2\lambda\mu}\mathbb{E}\|\tilde{y}_{t+1} - y_t\|^2 - \big(\frac{\gamma}{4} - \frac{25\gamma^3\kappa^2\tilde{L}^2}{\mu^2\lambda^2} - \frac{4\gamma^3\tilde{L}^2}{\mu\lambda}\big)\eta_t\|v_t\|^2 \\
&\quad + \frac{\gamma\alpha_{t+1}^2\sigma^2}{\lambda\mu\eta_t B} + \frac{\gamma\beta_{t+1}^2\sigma^2}{\lambda\mu\eta_t B} \\
&\leq -\frac{\gamma\tilde{L}^2\eta_t}{2}\mathbb{E}\|y_t - y^*(x_t)\|^2 - \frac{\gamma\eta_t}{2}\mathbb{E}\|\mathrm{grad}\Phi(x_t)\|^2 + \frac{\gamma\alpha_{t+1}^2\sigma^2}{\lambda\mu\eta_t B} + \frac{\gamma\beta_{t+1}^2\sigma^2}{\lambda\mu\eta_t B}, \quad (87)
\end{aligned}
$$

where the first inequality holds by Lemmas 5 and the above inequalities (83), (84) and (85); the second inequality is due to $\tilde{L} = \max(1, L_{11}, L_{12}, L_{21}, L_{22})$; the last inequality is due to $0 \leq \gamma \leq \frac{\mu\lambda}{2\kappa\tilde{L}\sqrt{25+4\mu\lambda}}$ and $\kappa \geq 1$.

According to the above inequality (87), we have

$$
\frac{\gamma\eta_t}{2}\big(\mathbb{E}\|\mathrm{grad}\Phi(x_t)\|^2 + \tilde{L}^2\mathbb{E}\|y_t - y^*(x_t)\|^2\big) \leq \Omega_t - \Omega_{t+1} + \frac{\gamma\alpha_{t+1}^2\sigma^2}{\lambda\mu\eta_t B} + \frac{\gamma\beta_{t+1}^2\sigma^2}{\lambda\mu\eta_t B}. \quad (88)
$$

Taking average over $t = 1, 2, \cdots, T$ on both sides of the inequality (88), we have

$$
\frac{1}{T}\sum_{t=1}^{T}\mathbb{E}\eta_t\big(\|\mathrm{grad}\Phi(x_t)\|^2 + \tilde{L}^2\|y_t - y^*(x_t)\|^2\big) \leq \sum_{t=1}^{T}\frac{2(\Omega_t - \Omega_{t+1})}{\gamma T} + \frac{1}{T}\sum_{t=1}^{T}\big(\frac{2\alpha_{t+1}^2\sigma^2}{\lambda\mu\eta_t B} + \frac{2\beta_{t+1}^2\sigma^2}{\lambda\mu\eta_t B}\big).
$$

Since the initial solution satisfies $y_1 = y^*(x_1) = \arg\max_{y\in\mathcal{Y}} f(x_1, y)$, we have

$$
\begin{aligned}
\Omega_1 &= \Phi(x_1) + \frac{6\gamma\tilde{L}^2}{\lambda\mu}\|y_1 - y^*(x_1)\|^2 + \frac{\gamma}{2\lambda\mu}\big(\frac{1}{\eta_0}\|\mathrm{grad}_x f(x_1, y_1) - v_1\|^2 + \frac{1}{\eta_0}\|\nabla_y f(x_1, y_1) - w_1\|^2\big) \\
&= \Phi(x_1) + \frac{\gamma}{2\lambda\mu}\big(\frac{1}{\eta_0}\|\mathrm{grad}_x f(x_1, y_1) - \mathrm{grad}_x f_{\mathcal{B}_1}(x_1, y_1)\|^2 + \frac{1}{\eta_0}\|\nabla_y f(x_1, y_1) - \nabla_y f_{\mathcal{B}_1}(x_1, y_1)\|^2\big) \\
&\leq \Phi(x_1) + \frac{\gamma\sigma^2}{\lambda\mu\eta_0 B}, \quad (89)
\end{aligned}
$$

where the last inequality holds by Assumption 5.

Table 3: Benchmark Datasets Used in Experiments

| datasets | #samples | #dimension | #classes |
|---|---|---|---|
| *MNIST* | 60,000 | $28 \times 28$ | 10 |
| *CIFAR-10* | 50,000 | $32 \times 32 \times 3$ | 10 |
| *CIFAR-100* | 50,000 | $32 \times 32 \times 3$ | 100 |
| *SVHN* | 73,257 | $32 \times 32 \times 3$ | 10 |
| *Fashion-MNIST* | 60,000 | $28 \times 28$ | 10 |
| *STL-10* | 5,000 | $32 \times 32 \times 3$ | 10 |

Consider $\eta_t$ is decreasing, i.e., $\eta_T^{-1} \geq \eta_t^{-1}$ for any $0 \leq t \leq T$, we have

$$\frac{1}{T} \sum_{t=1}^{T} \mathbb{E}\big(\|\text{grad}\Phi(x_t)\|^2 + \tilde{L}^2\|y_t - y^*(x_t)\|^2\big) \tag{90}$$

$$\leq \sum_{t=1}^{T} \frac{2(\Omega_t - \Omega_{t+1})}{T\gamma\eta_T} + \frac{1}{T\eta_T} \sum_{t=1}^{T} \Big(\frac{2\alpha_{t+1}^2\sigma^2}{\lambda\mu\eta_t B} + \frac{2\beta_{t+1}^2\sigma^2}{\lambda\mu\eta_t B}\Big)$$

$$\leq \frac{1}{T\eta_T}\Big(\frac{2\Phi(x_1)}{\gamma} + \frac{2\sigma^2}{\lambda\mu\eta_0 B} - \frac{2\Phi^*}{\gamma}\Big) + \frac{1}{T\eta_T}\sum_{t=1}^{T}\Big(\frac{2\alpha_{t+1}^2\sigma^2}{\lambda\mu\eta_t B} + \frac{2\beta_{t+1}^2\sigma^2}{\lambda\mu\eta_t B}\Big)$$

$$= \frac{2(\Phi(x_1) - \Phi^*)}{T\gamma\eta_T} + \frac{2\sigma^2}{T\lambda\mu\eta_0\eta_T B} + \frac{2(c_1^2 + c_2^2)\sigma^2}{T\eta_T\lambda\mu B}\sum_{t=1}^{T}\eta_t^3$$

$$\leq \frac{2(\Phi(x_1) - \Phi^*)}{T\gamma\eta_T} + \frac{2\sigma^2}{T\lambda\mu\eta_0\eta_T B} + \frac{2(c_1^2 + c_2^2)\sigma^2}{T\eta_T\lambda\mu B}\int_1^T \frac{b^3}{m+t}dt$$

$$\leq \frac{2(\Phi(x_1) - \Phi^*)}{T\gamma\eta_T} + \frac{2\sigma^2}{T\lambda\mu\eta_0\eta_T B} + \frac{2(c_1^2 + c_2^2)\sigma^2 b^3}{T\eta_T\lambda\mu B}\ln(m+T)$$

$$= \frac{2(\Phi(x_1) - \Phi^*)}{T\gamma b}(m+T)^{1/3} + \frac{2\sigma^2}{T\lambda\mu\eta_0 bB}(m+T)^{1/3} + \frac{2(c_1^2 + c_2^2)\sigma^2 b^2}{T\lambda\mu B}\ln(m+T)(m+T)^{1/3},$$

where the third inequality holds by $\sum_{t=1}^{T}\eta_t^3 \leq \int_1^T \eta_t^3 dt$. Let $M' = \frac{2(\Phi(x_1)-\Phi^*)}{\gamma b} + \frac{2\sigma^2}{\lambda\mu\eta_0 bB} + \frac{2(c_1^2+c_2^2)\sigma^2 b^2}{\lambda\mu B}\ln(m+T)$, we rewrite the above inequality as follows:

$$\frac{1}{T} \sum_{t=1}^{T} \mathbb{E}\big(\|\text{grad}\Phi(x_t)\|^2 + \tilde{L}^2\|y_t - y^*(x_t)\|^2\big) \leq \frac{M'}{T}(m+T)^{1/3}. \tag{91}$$

According to Jensen's inequality, we have

$$\frac{1}{T}\sum_{t=1}^{T}\mathbb{E}\big(\|\text{grad}\Phi(x_t)\| + \tilde{L}\|y_t - y^*(x_t)\|\big) \leq \Big(\frac{2}{T}\sum_{t=1}^{T}\mathbb{E}\big(\|\text{grad}\Phi(x_t)\|^2 + \tilde{L}^2\|y_t - y^*(x_t)\|^2\big)\Big)^{1/2}$$

$$\leq \frac{\sqrt{2M'}}{T^{1/2}}(m+T)^{1/6} \leq \frac{\sqrt{2M'}m^{1/6}}{T^{1/2}} + \frac{\sqrt{2M'}}{T^{1/3}}, \tag{92}$$

where the last inequality is due to $(a_1 + a_2)^{1/6} \leq a_1^{1/6} + a_2^{1/6}$ for all $a_1, a_2 > 0$.

$\square$

# B ADDITIONAL EXPERIMENTAL RESULTS

In this section, we provide additional experimental results on SVHN, FashionMNIST and STL-10 datasets, given in Table 3. The training loss and attack loss under uniform attack is shown in Fig 4. The test accuracy with natural images and uniform attack is shown in Tab. 4. From these results, our methods are robust to the uniform attack in training DNNs.

Table 4: Test accuracy against nature images and uniform attack for FashionMNIST, SVHN and STL-10 datasets.

| Method | Eval. Against | FashionMNIST | SVHN | STL-10 |
|--------|--------------|-------------|------|--------|
| RSGDA | Nat. Images | 84.96% | 75.72% | 52.34% |
| | Uniform Attack | **82.54%** | 43.19% | 47.28% |
| MVR-RSGDA | Nat. Images | 88.49% | 76.05% | 54.92% |
| | Uniform Attack | 76.75% | 45.06% | **48.89%** |
| SGDA | Nat. Images | **88.57%** | **91.96%** | **56.10%** |
| | Uniform Attack | 50.90% | **45.97%** | 45.27% |

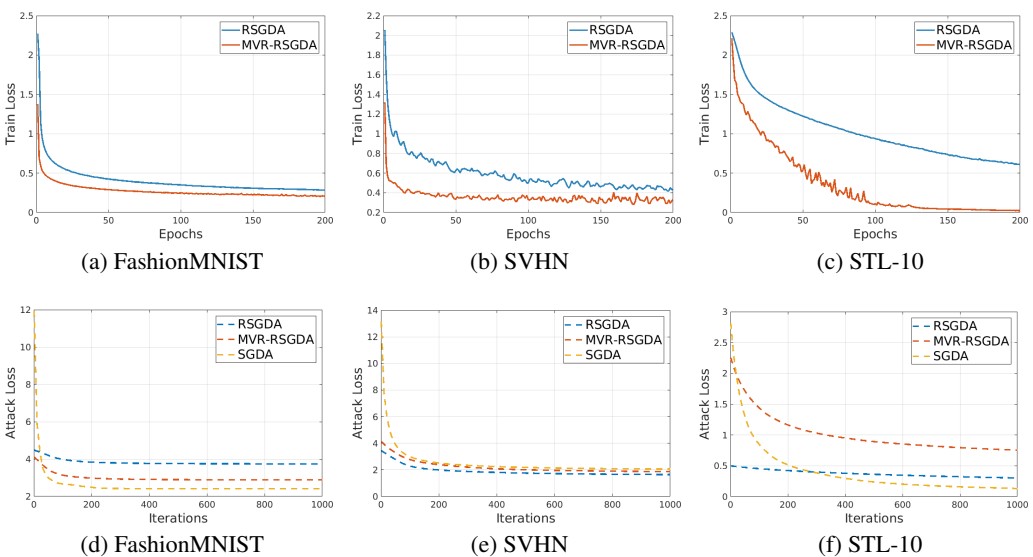

(a) FashionMNIST      (b) SVHN      (c) STL-10

(d) FashionMNIST      (e) SVHN      (f) STL-10

Figure 4: Additional results for robust training (a-c) and uniform attack (d-f) with SGDA, RSGDA and MVR-RSGDA algorithms.

