# OpenReview forum: "Gradient Descent Ascent for Min-Max Problems on Riemannian Manifolds"
_ICLR.cc/2021/Conference — Reject_

### Official Review · AnonReviewer4 · 2020-10-15
**The paper should be rejected due to the novelty issue**

**Rating:** 5
**Confidence:** 4

**Review:**

1.Summarize what the paper claims to do/contribute. Be positive and generous.

This paper considers solving the minimax saddle point of the form \min_{x\in X}\max_{y\in Y} f(x,y), where X is a Riemannian manifold and Y is a closed convex set. The objective function f is nonconvex in x and is strongly-concave in y. To the best of the reviewer’s knowledge, this min-max problem with Riemannian manifold constraint has not been considered in the literature. To solve such a problem, a two-time-scale manifold gradient descent ascent (MGDA) approach is then proposed to solve the problem in the deterministic case. An iteration complexity of O(\kappa^2\epsilon^{-2}) has been derived for the MGDA approach, where $\kappa$ is the condition number of the problem. When $f$ is in the form of an expectation or a finite-sum of a large number of component functions, the author proposes a manifold stochastic gradient descent ascent (MSGDA) approach and a variant of MSGDA with the gradient estimator constructed with a momentum-style SARAH/SPIDER technique. Sample complexities of O(\kappa^4\epsilon^{-4}) and \tilde O(\kappa^3\epsilon^{-3}) have been derived for the MSGDA and MVR-MSGDA methods respectively. The adversarial training of DNN and a specific instance of the DRO problem are proposed as the motivating example of this work.

2. Clearly state your decision (accept or reject) with one or two key reasons for this choice.

This paper should be rejected.

3. Provide supporting arguments for the reasons for the decision.

(i). The results for RGDA are largely from section 4 of (Tianyi Lin et al., 2020). Though the authors consider the Riemannian constraint on x, from the author’s experience in Riemannian optimization, it does not make much difference as long as the retraction operator is available.

(ii). The RSGDA analysis also shares the same novelty concern as stated in (i). Moreover, the sample complexity of (Tianyi Lin et al., 2020) is only O(\kappa^3\epsilon^{-4}), which is much better than the O(\kappa^4\epsilon^{-4}). The reviewer believes that following the proof of (Tianyi Lin et al., 2020) step by step while replacing the gradient step by gradient-retraction step, O(\kappa^3\epsilon^{-4}) will be achievable for RSGDA.

(iii). Technically, there are many non-rigorous arguments. E.g., the authors frequently use
\|x_{t+1} – x_t\|^2 \leq \gamma^2\eta_t^2\|v_t\|^2. However, this is wrong on manifold. The authors seems to believe \|R_x(u) - x\| = \|R_x(u) – R_x(0)\| \leq \|u\|, but this is not true. The rigorous statement should be: Exist d,c>0 s.t. for any x\in X, for any u\inT_xX with \|u\|\leq d, we have \|R_x(u) - x\| \leq c\|u\|. These parameters should present in the final complexity result. They represent the property of the manifold itself.

(iv). Although the result for momentum SARAH/SPIDER variance reduction seems new in this setting, merely this part is not enough contribution.

4. Provide additional feedback with the aim to improve the paper. Make it clear that these points are here to help, and not necessarily part of your decision assessment.

(I) The authors should carefully describe the relation of their result and that of (Tianyi Lin et al., 2020). Specifically, clearly state the differences and additional difficulties when generalizing the result of (Tianyi Lin et al., 2020).

(II) The reviewer suspect that the result of RSGDA can be improved to O(\kappa^3\epsilon^{-4}) by following the procedure of (Tianyi Lin et al., 2020) more carefully. If this can be done, it will be better fill this gap. If this cannot be done, the authors should state the difficulty.

(III) Since this work generalizes (Tianyi Lin et al., 2020), and the Riemannian manifold does not add much difficulty. So it might be better to include the nonconvex-concave case in (Tianyi Lin et al., 2020), this will make the result more complete.

(IV) The authors should carefully review their statements w.r.t. Riemannian manifold so that the mistakes pointed out before does not happen. Though, generalizing from Euclidean optimization to Riemannian optimization is conceptually simple. But there are also some subtly in the arguments.

---

> ### Author Response · Authors · 2020-11-23
> **Authors' responses**
>
> Thanks for your constructive comments.
>
>
> (i): ...
>
> A4-1: Thanks for your valuable comment. We first declare the our main results are not derived from  section 4 of (Tianyi Lin et al., 2020). There exist many differences between our paper with (Tianyi Lin et al., 2020) as follows:
>
> 1) In the algorithms, besides Riemannian optimization operators such as retraction,  when updating parameter y, our algorithms not only use a projection operator but also use a convex combination, while (Tianyi Lin et al., 2020) only use a projection operator. Please see the step 6 in our Algorithms 1 and 2.
>
> 2) In the theoretical analysis, we only assume the convexity of constraint set $\mathcal{Y}$, while (Tianyi Lin et al., 2020) and (Tianyi Lin et al., 2020b) not only assume the convexity of set $\mathcal{Y}$, but also assume and use its bounded (Please see Assumption 4.2 in (Tianyi Lin et al., 2020)). Clearly, our assumption is milder than (Tianyi Lin et al., 2020) and (Tianyi Lin et al., 2020b). When there does not exist a constraint set on parameter $y$, i.e.,$\mathcal{Y}=R^d$, our algorithms and theoretical results still work, while (Tianyi Lin et al., 2020)  and  (Tianyi Lin et al., 2020b) can’t work.
>
> 3) In the theoretical analysis, we use a new convergence metric $H_t=\|\nabla\Phi(x_t)\| + \tilde{L}\|y_t-y^*(x_t)\|$ (please see the equality (10) in our paper). In fact, the metric $\|\nabla\Phi(x_t)\|$ used in (Tianyi Lin et al., 2020) is only a part of our metric. Clearly, our metric is larger than that of (Tianyi Lin et al., 2020). Thus, our theoretical results are  tighter  than (Tianyi Lin et al., 2020).
>
> 4) In fact, our paper mainly focuses on the stochastic min-max problem, and the fast variance-reduced MVR-RSGDA Algorithm is our main contribution. In our MVR-RSGDA Algorithm, the stochastic gradients $v_{t+1}$ and $w_{t+1}$ are biased.  While the SGDA algorithm in (Tianyi Lin et al., 2020) relies on unbiased stochastic gradient. Clearly, the algorithms of (Tianyi Lin et al., 2020) can not easily extend our MVR-RSGDA Algorithm. In addition, how to choose the learning rate $\eta_t$ and parameters $ \alpha_t$, $\\beta_t$ that can not be derived from the section 4 of (Tianyi Lin et al., 2020).
>
> (Tianyi Lin et al., 2020)  On Gradient Descent Ascent for Nonconvex-Concave Minimax Problems, ICML, 2020. https://arxiv.org/pdf/1906.00331.pdf
>
> (Tianyi Lin et al., 2020b) Near-Optimal Algorithms for Minimax Optimization, COLT 2020. https://arxiv.org/pdf/2002.02417.pdf
>
> (ii): ...
>
> A4-2: Thanks for your valuable suggest. From the above responses in A4-1, the sample complexity of  (Tianyi Lin et al., 2020)  depends on some more strict assumptions than our assumptions. Our RSGDA algorithm may be obtain a better sample complexity depended on some strict assumptions as in (Tianyi Lin et al., 2020).
>
> (iii): ...
>
> A4-3: Thanks for your comment. We first declare that we don’t use the inequality $|x_{t+1} – x_t|^2 \leq \gamma^2\eta_t^2|v_t|^2$, and it never appeared in my paper.
> We guess from what you mean, you think that the inequalities (47) and (74) in our paper use the inequalities $|x_{t+1} – x_t|^2 \leq \gamma^2\eta_t^2|v_t|^2$. In fact, this conclusion is wrong. You get this conclusion completely from the simple Euclidean view instead of manifold view.
>
> Specifically, the inequality (47) is due to our Lemma 4, which is dependent on Assumption 1. In Assumption 1, the parameters $L_{11}$ and $L_{21}$ implicitly includes curvature information as in (Han & Gao,2020). Similarly, the inequality (74) is due to Lemma 1. We emphasize that the smoothness of $f(x,y)$ over manifold $M$ is retraction smooth.
> We welcome you to point out other places, which maybe use this inequality.
>
> (iv)...
>
> A4-4: In fact, our paper mainly focuses on the stochastic min-max problem, and the fast variance-reduced MVR-RSGDA Algorithm is our main contribution. From our MVR-RSGDA Algorithm, the stochastic gradients $v_{t+1}$ and $w_{t+1}$ are biased.  While the SGDA algorithm in (Tianyi Lin et al., 2020) relies on unbiased stochastic gradient. Clearly, the algorithms of (Tianyi Lin et al., 2020) can not easily extend our MVR-RSGDA Algorithm.
> Moreover, our MVR-RSGA algorithm reaches a lower sample complexity without relying on large-scale mini-batches. While the SGDA algorithm not only suffers from high sample complexity, but also relies on large mini-bathes.
>
> (I)...
>
> A4-5:  Please see the above A4-1.
>
> (II)...
>
> A4-6:  Thanks for your suggestion. I think that our RSGDA algorithm may also obtain an improved sample complexity based similar assumptions in (Tianyi Lin et al., 2020).
>
> (III)...
>
> A4-7: Thanks for your suggestion. We will extend our algorithms to the nonconvex-concave case. We still emphasize that our  MVR-RSGDA Algorithm can not be derived from (Tianyi Lin et al., 2020), please see the above A4-1.
>
> (IV)...
>
> A4-8: Please see the above A4-3.

---

### Official Review · AnonReviewer2 · 2020-10-22

**Rating:** 4
**Confidence:** 4

**Review:**

In this paper, the authors present and analyze a class of gradient-descent algorithms for solving min-max problems when the first (minimization) variable is constrained to live on a Riemaniann manifold. In the case when i) a retraction and an isometric transport are available on the manifold; and ii) the objective is strongly convex and smooth in the second variable, the authors show convergence rates. Experiments are performed with the setting of minimizing losses of neural nets whose weights are constrained to live in the Stiefeld manifold while an attacker of small norm perturbs the input.


This work is one of the first I see tackling explicitly min-max optimization over a Riemannian manifold. The presented techniques are interseting and seem to have some applications. However, the presentation and writing of the make it quite hard to follow and, in my opinion, not ready for publication at that stage.


* The introduction is rather well written and nice to read.

* Then, there is an imprecision that has been bothering me throughout the paper: the authors seem to actually define the nabla_x f(x,y) directly as the Riemannian gradient, not the usual gradient. This is done directly without mentioning it in the bottom of page 3 (otherwise, all the gradient transportations do not make sense). Considering how the functions are defined in eg. (3,4), nabla_x f could stand for the full gradient naturally and then, grad_x f = proj T_x (nabla_x  f ) could be the Riemannian gradient. I think this can be easily fixed but presently it can cause a lot of confusion.
- In addition, in Assuùmption 1, I do not get the inequality with L21. nabla_y lives in R^n so the transport is identity?...


* The authors state that "this is the first study of minmax optimization over the Riemannian manifold" however, there seem to be some works on finding stable points of variational inequalities (by extragradient and the like) on Manifolds, eg.
- (see Sec. 5.2 ) Li, Chong, Genaro López, and Victoria Martín-Márquez. "Monotone vector fields and the proximal point algorithm on Hadamard manifolds." Journal of the London Mathematical Society 79.3 (2009): 663-683.
- Wang, J. H., et al. "Monotone and accretive vector fields on Riemannian manifolds." Journal of optimization theory and applications 146.3 (2010): 691-708.
- Ferreira, Orizon Pereira, LR Lucambio Pérez, and Sandor Z. Németh. "Singularities of monotone vector fields and an extragradient-type algorithm." Journal of Global Optimization 31.1 (2005): 133-151.

* In assumption 1, the terms L11 and L12 seem to implicitly contain curvature information on the manifold. Could the authors comment on that? I guess that Assumptions 2 may also imply some "hidden" conditions on Y.

* Theorem 1: In the case of deterministic gradients ( ie (8) ), I do not get over what the expectation is. Either it is related to the gradient and hence not need. Or, it is related to the random output (line 10) in which  case the sum seems not needed (as in E[H_\zeta] below).

* In Table 2, How are the algorithms stopped, is it after 1000 iterations?

* I do not get how the authors obtain the first line after "By plugging the inequalities (22) into (25), we have" in Appendix A


Minor Comments/Typos:
* Abstract: "on the Riemannian manifold" -> "on Riemannian manifolds"
* Preliminairies: several typos "We define a retraction R maps tangent space" "Exp mapping is a special case of retraction that approximate the Exp mapping up the first order" (!)
* Theorem 1: "Suppose the sequence () be generated"
* In Remark 2, the setting B=T seems quite unrealistic.
* Figure 3: "Iteratons"->"Iterations" in the 3 subfigures.
* between (22) and (23) "By the function f is strongly concave"

---

> ### Author Response · Authors · 2020-11-23
> **Authors' responses**
>
> Thanks for your constructive comments.
>
> Q1: Then, there is an imprecision that has been bothering me throughout the paper: the authors seem to actually define the nabla_x f(x,y) directly as the Riemannian gradient, not the usual gradient. This is done directly without mentioning it in the bottom of page 3 (otherwise, all the gradient transportations do not make sense). Considering how the functions are defined in eg. (3,4), nabla_x f could stand for the full gradient naturally and then, grad_x f = proj T_x (nabla_x f ) could be the Riemannian gradient. I think this can be easily fixed but presently it can cause a lot of confusion.
>
> A3-1: Thanks for your suggestion. We let $grad_x f$ denote the Riemannian gradient in our revised manuscript.
>
> Q2: In addition, in Assumption 1, I do not get the inequality with $L_{21}$. $\nabla_y$ lives in $R^n$ so the transport is identity?...
>
> A3-2: In Assumption 1, gradient $grad_y f(x_1,y;\xi)$ is on tangent space $T_{x_1}\mathcal{M}$, and gradient $grad_y f(x_2,y;\xi)$ is on tangent space $T_{x_2}\mathcal{M}$. Please see Assumption 1 and remark about it in our revised paper.
>
> Q3: The authors state that "this is the first study of minmax optimization over the Riemannian manifold" however, there seem to be some works on finding stable points of variational inequalities (by extragradient and the like) on Manifolds.
>
> A3-3: Thanks for you valuable comment. In our revised paper , we delete this statement, and introduce these works in the related work part of our paper.
>
> Q4: In assumption 1, the terms $L_{11}$ and $L_{12}$ seem to implicitly contain curvature information on the manifold. Could the authors comment on that? I guess that Assumptions 2 may also imply some "hidden" conditions on Y.
>
> A3-4: In assumption 1, the term $L_{11}$ and $L_{21}$ implicitly contain the curvature information as in (Han & Gao,2020) and (Han & Gao,2020b). In fact, Assumption 2 does not imply the condition on $Y$, and only imposes the retraction smooth of function $\Phi(x)$.
>
>
> (Han & Gao,2020). Riemannian stochastic recursive momentum method for non-convex optimization. arXiv353 preprint arXiv:2008.04555 .
>
> (Han & Gao,2020b). Variance reduction for Riemannian non-convex optimization with batch size
> adaptation. arXiv preprint arXiv:2007.01494 .
>
> Q5: Theorem 1: In the case of deterministic gradients ( ie (8) ), I do not get over what the expectation is. Either it is related to the gradient and hence not need. Or, it is related to the random output (line 10) in which case the sum seems not needed (as in E[H_\zeta] below).
>
> A3-5: Thanks for your comment. Yes, the expectation in Theorem 1 is not required. We will delete it.
>
> Q6:In Table 2, How are the algorithms stopped, is it after 1000 iterations?
>
> A3-6: Yes, they are stopped after 1000 iterations.
>
> Q7: I do not get how the authors obtain the first line after "By plugging the inequalities (22) into (25), we have" in Appendix A.
>
> A3-7:  Sorry for a clerical error in the inequality (21) resulting in a mistake in the inequality (22). We correct these inequalities as follows:
> the correct inequality (21):
> \begin{align}
>   (y-y^*(x_2))^T\nabla_y f(x_2,y^*(x_2)) \leq 0, \quad \forall y\in \mathcal{Y}.
> \end{align}
>
> the correct inequality (22):
> \begin{align}
>   (y^*(x_2)-y^*(x_1))^T\big(\nabla_y f(x_1,y^*(x_1)) - \nabla_y f(x_2,y^*(x_2))\big) \leq 0.
>  \end{align}
>
> Based on the correct inequality (22), we can get the inequality (26).
> Please check our revised manuscript.
>
>
> Q8: In Remark 2, the setting B=T seems quite unrealistic.
>
> A3-8: Thanks for your comment. Yes, this setting $B=T$ is unrealistic in the RSGDA Algorithm. Thus, we further propose a more efficient MVR-RSGDA algorithm without large batches, which reaches the best known sample complexity for its Euclidean counterparts.
>
>
> Thanks for your minor comments.   We correct these minor issues in our revised paper.

---

### Official Review · AnonReviewer3 · 2020-10-28
**Significant mismatch between theory and experiments; incremental theory.**

**Rating:** 4
**Confidence:** 3

**Review:**

This paper studies the convergence of min-max optimization algorithms when the minimizer takes values in a manifold and the maximizer in a subset of R^d. The authors proposed three algorithms and analyzed the convergence rates to stationary points. Some experiments are provided to evaluate the proposed algorithms.

There are two major issues with the paper:

1) There is essentially no novelty in the theory. Specifically, under the very strong assumptions 1-5 and that the maxmization is strongly concave, one can simply pretend that the retraction is a gradient step and then the analysis for gradient descent-ascent would hold almost verbatim for the proof provided by the authors. In other words, as the difficulty of analyzing Riemannian algorithms is already "assumed away" by the assumptions, there is virtually zero element of Riemannian geometry in the proofs. As a result, I found the theory to be incremental at best.


2) The empirical evaluation is rather incomplete and disconnected from the theory:

2a) First, the only algorithms presented are the ones proposed in this paper. Where are the baselines? What is the retraction used and what about its computational efficiency?

2b) The main motivation (and the only considered applications) of the paper is robust training. However, the robust training problem is more commonly formulated as min-\sum-max instead of min-max-\sum, and there is a significant difference in optimizing these two objectives. The authors did not provide any justification as why they could switch the order of \sum and max at will.


Based on the above, I recommend rejection.

---

> ### Author Response · Authors · 2020-11-23
> **Authors' responses**
>
> Thanks for your constructive comments.
>
> Q1.There is essentially no novelty in the theory. Specifically, under the very strong assumptions 1-5 and that the maxmization is strongly concave, one can simply pretend that the retraction is a gradient step and then the analysis for gradient descent-ascent would hold almost verbatim for the proof provided by the authors....
>
> A2-1: Thanks for your comment. We first declare that our Assumptions 1-5 are very common in Riemannian optimization, min-max optimization and stochastic optimization.
> Specifically, Assumption 1 imposes the retraction smooth over manifold and the smooth over Euclidean space, respectively, which are commonly used in Riemannian optimization (Han & Gao,2020), (Han & Gao,2020b) ,(Sato et al. 2019), and min-max optimization (Tianyi Lin et al., 2020). The term$ L_{11}$, $L_{12}$ and $L_{21}$ implicitly contain the curvature information as in (Han & Gao,2020) and (Han & Gao,2020b). Assumption 2 imposes the retraction smooth of function $ \Phi(x) $, as in (Han & Gao,2020). Assumption 3 imposes the strongly concave of $f(x,y)$ on variable $y$, as in (Tianyi Lin et al., 2020) (Luo Luo, et al.,2020). Assumption 4 guarantees the feasibility of the nonconvex-strongly-concave problems, as in (Tianyi Lin et al., 2020) (Luo Luo, et al.,2020). Assumption 5 imposes the bounded variance of stochastic (Riemannian) gradients, which is commonly used in the stochastic optimization (Tianyi Lin et al., 2020) (Luo Luo, et al.,2020).
>
> We also declare that our theoretical results are not derived from the exiting min-max optimization such as (Tianyi Lin et al., 2020). There exist some reasons as follows:
>
> 1-1)  In the algorithms, besides Riemannian optimization operators such as retraction,  when updating parameter y, our algorithms not only use a projection operator but also use a convex combination, while (Tianyi Lin et al., 2020) only use a projection operator. Please see the step 6 in our Algorithms 1 and 2.
>
> 1-2) In the theoretical analysis, we only assume the convexity of constraint set $\mathcal{Y}$, while (Tianyi Lin et al., 2020) and (Tianyi Lin et al., 2020b) not only assume the convexity of set $\mathcal{Y}$, but also assume and use its bounded (Please see Assumption 4.2 in (Tianyi Lin et al., 2020)). Clearly, our assumption is milder than (Tianyi Lin et al., 2020) and (Tianyi Lin et al., 2020b). When there does not exist a constraint set on parameter y, i.e.,$\mathcal{Y}=R^d$, our algorithms and theoretical results still work, while (Tianyi Lin et al., 2020)  and  (Tianyi Lin et al., 2020b) can’t work.
>
> 1-3) In the theoretical analysis, we use a new convergence metric $H_t=\|\nabla\Phi(x_t)\| + \tilde{L}\|y_t-y^*(x_t)\|$ (please see the equality (10) in our paper). In fact, the metric $\|\nabla\Phi(x_t)\|$ used in (Tianyi Lin et al., 2020) is only a part of our metric. Clearly, our metric is larger than that of (Tianyi Lin et al., 2020). Thus, our theoretical results are  tighter  than (Tianyi Lin et al., 2020).
>
> 1-4)	 In fact, our paper mainly focuses on the stochastic min-max problem, and the fast variance-reduced MVR-RSGDA Algorithm is our main contribution. In MVR-RSGDA Algorithm, the stochastic gradients $v_{t+1}$ and $w_{t+1}$ are biased.  While the SGDA algorithm in (Tianyi Lin et al., 2020) relies on unbiased stochastic gradient. Clearly, the algorithms of (Tianyi Lin et al., 2020) can not easily extend our MVR-RSGDA Algorithm.
>
> (Luo Luo, et al.,2020) Stochastic recursive gradient descent ascent for stochastic nonconvex-strongly-concave minimax problems. NeurIPS, 2020.
>
>
>
> Q2.The empirical evaluation is rather incomplete and disconnected from the theory:
>
> 2a) First, the only algorithms presented are the ones proposed in this paper. Where are the baselines? What is the retraction used and what about its computational efficiency?
>
> A2-2: We added the SGDA algorithm (Tianyi Lin et al., 2020)  as a baseline, which does not apply the orthogonal regularization in DNN robust training. Please see Fig.3 and Tab.2. QR decomposition is used as the retraction function. The computational cost is similar to standard QR.
>
> 2b) The main motivation (and the only considered applications) of the paper is robust training. However, the robust training problem is more commonly formulated as min-\sum-max instead of min-max-\sum,....
>
> A2-3：Thanks for your comment. In our paper, we only consider a special case of robust training, where we estimate a universal perturbation for all samples as in (Moosavi-Dezfooli et al., 2017) and ( Chaubey et al., 2020). Under this case, the robust training can be formulated as min-max-\sum. Please see the equality (2).
>
> (Moosavi-Dezfooli et al., 2017). Universal adversarial perturbations. In Proceedings of the IEEE conference on computer vision and pattern recognition, pp. 1765–1773, 2017.
>
> ( Chaubey et al., 2020). Universal adversarial perturbations: A survey. arXiv preprint arXiv:2005.08087, 2020.

---

### Official Review · AnonReviewer1 · 2020-11-10
**The paper focuses on an important class of problems**

**Rating:** 7
**Confidence:** 4

**Review:**

The paper proposes Riemannian algorithms for min-max problems where the min problem is over a manifold and the max problem is a strongly convex problem. It presents a rigorous convergence analysis of the algorithms proposed. Finally, experiments show the good performance of the algorithms for robust training of DNNs over manifolds.

The paper is nicely written and overall easy to follow. The work is relevant for practitioners who use similar min-max problems as it provides necessary theoretical backing to the use of such constraints in DNNs.

Comments
Assumptions on the manifold are missing, compactness is required? What are other restrictions?

Since the max problem is strongly convex, one can in principle directly work the formulation min_{x in M} Phi(x). The Euclidean gradient of Phi(x) can be easily computed (Danskin’s theorem). (One work in a similar spirit is [1]). Of course, I assume that the max problem is solvable with an iterative solver which is easy to implement in many cases. This may be easier to analyze, no? If that is so, then why go for explicit updates of the max variable y? A discussion on this is missing.

The main contributions are theoretical, and to that end, the experiments suffice. However, it can be broadened by including other baselines. This will strengthen the paper.

There are some issues with how to use \citet and \citep while citing papers, especially in Section 2.2. For example, in “More recently, Sato et al. (2019) has proposed…,” it should either be \citep or has ---> have. Please look into other such issues in the paper.

[1] Jawanpuria, P. and Mishra, B., 2018, July. A unified framework for structured low-rank matrix learning. In International Conference on Machine Learning (pp. 2254-2263).

---

> ### Author Response · Authors · 2020-11-23
> **Authors' responses**
>
> Thanks for your constructive comments.
>
> Q1.	Comments Assumptions on the manifold are missing, compactness is required? What are other restrictions?
>
> A1-1: Thanks for your comment. Our methods are not required to impose the compactness of the whole manifold $\mathcal{M}$. In fact, as in (Sato et al. 2019) (Han & Gao,2020) (Han & Gao,2020b), our methods only need to assume that the iterate sequences generated from algorithms stay continuously in a neighbourhood $\mathcal{X}\subset \mathcal{M}$ around an optimal point $x^*$. And $\mathcal{X}$ is a compact set. We have added this assumption in the revised manuscript.
>
> (Han & Gao,2020). Riemannian stochastic recursive momentum method for non-convex optimization. arXiv353 preprint arXiv:2008.04555 .
>
> (Han & Gao,2020b). Variance reduction for Riemannian non-convex optimization with batch size
> adaptation. arXiv preprint arXiv:2007.01494 .
>
> (Sato et al. 2019). Riemannian stochastic variance reduced gradient algorithm with retraction and vector transport. SIAM Journal on Optimization, 29(2):1444–1472, 2019.
>
>
> Q2.Since the max problem is strongly convex, one can in principle directly work the formulation min_{x in M} Phi(x). The Euclidean gradient of Phi(x) can be easily computed (Danskin’s theorem). (One work in a similar spirit is [1]). Of course, I assume that the max problem is solvable with an iterative solver which is easy to implement in many cases. This may be easier to analyze, no? If that is so, then why go for explicit updates of the max variable y? A discussion on this is missing.
>
> A1-2: In fact, following (Tianyi Lin et al., 2020), it is more effective and efficient to alternately solve the min-max problem (like as Gauss–Seidel method)  than to solve the max problem first and then solve the mini problem (like as Jacobi method).  Yes, this iterative solver should be easily analyzed. We will discuss this in the final manuscript.
>
> (Tianyi Lin et al., 2020)  On Gradient Descent Ascent for Nonconvex-Concave Minimax Problems, ICML, 2020. https://arxiv.org/pdf/1906.00331.pdf
>
> Q3.The main contributions are theoretical, and to that end, the experiments suffice. However, it can be broadened by including other baselines. This will strengthen the paper.
>
> A1-3: Thanks for your suggestion. We added the SGDA algorithm (Tianyi Lin et al., 2020)  as a baseline, which does not apply the orthogonal regularization in DNN robust training. Please see Fig.3 and Table.2 in the revised manuscript.
>
>
> Thanks for your minor comments about citing papers. We correct it in our revised manuscript.

---

### Decision · Program_Chairs · 2021-01-07
**Final Decision**

**Decision:**

Reject

**Comment:**

In this paper, the authors provide a Riemannian version of gradient descent/ascent for min-max problems on manifolds. Assuming a tractable retraction mapping for the descent/ascent step, the authors provide a complexity analysis for finding a (local) saddle point in the spirit of Lin et al. (2020).

The paper received three negative recommendations and one positive, with all reviewers indicating high to expert confidence. After my own reading of the paper, I concur with the majority view that the paper does not clear the (admittedly high) bar for ICLR. After discussing the paper with the reviewers, the concerns that led to this recommendation are as follows:
1. On the proposed examples: it is not clear what exactly is the motivation for the DNN training example with orthonormality constraints on the weights. In the paper, $x$ is the vector of DNN weights, so it lives in some real space $R^d$. The authors subsequently assume it is constrained to live on some Stiefel manifold, but which one? The Stiefel manifold is the set of all orthonormal $r$-frames on $R^d$, and the formulation in Section 6 doesn't clarify things. Moreover, the papers cited by the authors either concern the initialization of the DNN or a regularization by an orthonormality/orthogonality penalty. This is quite different since DNN training typically involves at least some neurons with very large weights (which is of course disallowed if the weights are constrained to live on some "norm 1" subspace).
2. The presentation is often lacking in mathematical rigor: in a Riemannian setting, it is crucial to distinguish between the Riemannian distance function and the Riemannian norm. However, the two were used interchangeably at several points, and the authors' revision wasn't satisfactory in this regard - even the basepoint for the norm is missing in cases where it is not made clear from the context on which point the norm is considered.
3. The cost of applying retraction-based methods in DNNs is also unclear. On a Stiefel manifold, the only known retractions involve SVD, so they have a superlinear computational cost relative to the size of the input matrix. If the dimensionality of the input matrix is that of the parameter space of the DNN, the efficiency of the method seems somewhat limited.

The above concerns regarding DNN training are perhaps less important if we view this as a primarily theoretical paper. In that regard however, the novelty of this paper over that of Lin et al. (and other Riemannian minimization papers) is not clear, so I am forced to recommend rejection at this stage. That being said, I believe that a thoroughly revised version of this paper could ultimately be publishable at one of the top venues of the field, and I would strongly encourage the authors to pursue this.